# GRAND-SLAMIN' Interpretable Additive Modeling with Structural Constraints

**Shibal Ibrahim**
MIT
Cambridge, MA
shibal@mit.edu

**Gabriel Isaac Afriat**
MIT
Cambridge, MA
afriatg@mit.edu

**Kayhan Behdin**
MIT
Cambridge, MA
behdink@mit.edu

**Rahul Mazumder**
MIT
Cambridge, MA
rahulmaz@mit.edu

## Abstract

Generalized Additive Models (GAMs) are a family of flexible and interpretable models with old roots in statistics. GAMs are often used with pairwise interactions to improve model accuracy while still retaining flexibility and interpretability but lead to computational challenges as we are dealing with order of $p^2$ terms. It is desirable to restrict the number of components (i.e., encourage sparsity) for easier interpretability, and better computational and statistical properties. Earlier approaches, considering sparse pairwise interactions, have limited scalability, especially when imposing additional structural interpretability constraints. We propose a flexible GRAND-SLAMIN framework that can learn GAMs with interactions under sparsity and additional structural constraints in a differentiable end-to-end fashion. We customize first-order gradient-based optimization to perform sparse backpropagation to exploit sparsity in additive effects for any differentiable loss function in a GPU-compatible manner. Additionally, we establish novel non-asymptotic prediction bounds for our estimators with tree-based shape functions. Numerical experiments on real-world datasets show that our toolkit performs favorably in terms of performance, variable selection and scalability when compared with popular toolkits to fit GAMs with interactions. Our work expands the landscape of interpretable modeling while maintaining prediction accuracy competitive with non-interpretable black-box models. Our code is available at https://github.com/mazumder-lab/grandslamin.

## 1 Introduction

Many state-of-the-art learners e.g., tree ensembles, neural networks, kernel support vector machines, can be difficult to interpret. There have been various efforts to derive some post-training explainability from these models —see [5] for a survey. Post-hoc explainability attempts to explain black-box prediction with interpretable instance-specific approximations e.g, LIME [49] and SHAP [35]. However, such approximations are known to be unstable [12, 28], expensive [51] and inaccurate [32]. Hence, it is desirable to consider modeling approaches that are inherently interpretable.

Amongst classical approaches, there are some models that have inherent interpretability e.g., Linear models, CART [4] and Generalized Additive Models (GAMs)[14]. GAMs [15] which have old roots in statistics are considered a front-runner in the context of interpretable modeling. They consider an additive model of the main effects of the form: $g(\mathbb{E}[y]) = \sum_{j \in [p]} f_j(x_j)$, where $x_j$ denotes the $j$th feature in input $\boldsymbol{x} \in \mathbb{R}^p$, $f_j$ is a univariate shape function and $g$ denotes the link function that adapts the model to various settings such as regression or classification. GAMs are considered easy to interpret as the impact of each feature can be understood via visualizing the corresponding shape function e.g., plotting $f_j(x_j)$ vs $x_j$. However, such models often suffer in prediction performance when compared to black-box methods e.g., deep neural networks (DNNs). This can be attributed in part to the fact that GAMs do not consider interactions among covariates.

37th Conference on Neural Information Processing Systems (NeurIPS 2023).

There has been some exciting work that aims to reduce this performance gap by considering GAMs with pairwise interactions [see, for example, 34, 60, 6, 46, 9, 10, and the references therein]. GAMs with pairwise interactions consider a model of the following form:

$$g(\mathbb{E}[y]) = \sum_{j \in [p]} f_j(x_j) + \sum_{(j,k) \in \mathcal{I}} f_{j,k}(x_j, x_k) \tag{1}$$

where $f_{j,k}$ is a bivariate shape function and $\mathcal{I} \subseteq \{(1,2),(1,3),...,(p-1,p)\}$ denotes the set of all pairwise interactions. Under this model, $f_j(x_j)$ is the $j$-main effect and $f_{j,k}(x_j, x_k)$ is the $(j,k)$-th interaction effect. Pairwise interactions are considered interpretable as each of the bivariate shape function $f_{j,k}$ can be visualized as a heatmap on an $x_j, x_k$-plane. Despite their appeal, GAMs with pairwise interactions pose several challenges: (i) Learning all pairwise interaction effects of the order of $p^2$ lead to computational and statistical challenges. (ii) Performing component selection such that only a few of the components $\{f_j\}$ and $\{f_{j,k}\}$ are nonzero in an end-to-end fashion (while training) is a hard combinatorial optimization problem. We remind the reader that component selection is needed to aid interpretability. (iii) Imposing structural constraints on the interaction effects, e.g., hierarchy [42, 13, 7, 39, 3, 60, 10] makes the associated optimization task more complex.

In this paper, we introduce a novel `GRAND-SLAMIN` framework that allows for a flexible way to do component selection in GAMs with interactions under additional structural constraints in an end-to-end fashion. In particular, we introduce an alternative formulation of GAMs with interactions with additional binary variables. Next, we *smooth* these binary variables so that we can effectively learn these components via first-order methods in smooth optimization (e.g, SGD). Our formulation appears to have an edge over existing methods in terms of (i) model flexibility for enhanced structural interpretability and (ii) computational efficiency. First, the binary variables allow us to impose in the model (a) component selection constraints and (b) additional structural constraints (e.g, hierarchy) via a unified optimization formulation. Both of constraints (a), (b) can aid interpretability, model compression, and result in faster inference and better statistical properties. Second, the our smoothing procedure for the binary variables allows us to have customized algorithms that exploit sparsity in the forward and backward pass of the backpropagation algorithm.

For structural interpretability, we study two notions: weak and strong hierarchy [42, 13, 7, 39, 3].

$$\text{Weak Hierarchy}: \quad f_{j,k} \neq 0 \implies f_j \neq 0 \quad \text{or} \quad f_k \neq 0 \quad \forall (j,k) \in \mathcal{I}, \ j \in [p], \ k \in [p]. \tag{2}$$
$$\text{Strong Hierarchy}: \quad f_{j,k} \neq 0 \implies f_j \neq 0 \quad \text{and} \quad f_k \neq 0 \quad \forall (j,k) \in \mathcal{I}, \ j \in [p], \ k \in [p]. \tag{3}$$

Weak hierarchy allows an interaction effect $f_{j,k}$ to be selected if either main effect $f_j$ or $f_k$ is selected. Strong hierarchy allows for an interaction effect $f_{j,k}$ to be selected only if both main effects $f_j$ and $f_k$ are selected. Such hierarchy constraints are popular in high-dimensional statistics: (i) They lead to more interpretable models [36, 3, 10], (ii) They promote practical sparsity, i.e., reduce the number of features that need to be measured when making new predictions (see Sec. 6.2) — this can reduce future data collection costs [3, 61], (iii) Additional constraints can also help regularize a model, sometimes resulting in improved AUC (see Sec. 6.1). (iv) They can also reduce variance in estimation of main effects in the presence of interaction effects (see Sec. 6.4), allowing the user to have more "trust" on model interpretability explanations.

**Contributions.** To summarise, while it's well acknowledged that GAMs with sparse interactions are a useful flexible family of explainable models, learning them pose significant computational challenges due to the combinatorial nature of the associated optimization problem. Our technical contributions in this paper be summarized as:

1. We propose a novel optimization formulation that makes use of indicator (binary) variables. The indicator variables allow us to impose both (a) component selection and (b) structural constraints in an end-to-end fashion. We consider a smooth and continuous parameterization of the binary variables so that the optimization objective is differentiable (for a smooth training loss) and hence amenable to first order methods such as SGD.
2. We show the flexibility of our framework by considering two different notions of hierarchy. While these constraints improve interpretability, they make the combinatorial problem more challenging [3, 17]. We propose end-to-end algorithms to train these models, making our approach quite different from existing neural-based toolkits [60, 10].
3. We exploit sparsity in the indicator variables during the course of the training for sparse forward and backward passes in a customized backpropagation algorithm in a GPU-compatible manner. This provides speedups on training times up to a factor of $10\times$ over standard backpropagation.

4. We study novel statistical properties of our model, and present non-asymptotic prediction error bounds. Different from earlier work, our results apply to learning with tree-based shape functions (for both main and interaction effects).
5. We introduce a new open-source toolkit `GRAND-SLAMIN` and perform experiments on a collection of 16 real-world datasets to demonstrate the effectiveness of our toolkit in terms of prediction performance, variable selection and scalability.

## 2    Related Work

**GAMs.** GAMs have a long history in statistics [14] and have been extensively studied. They're often studied with smooth spline shape functions [see, e.g, 37, 47, 20, 63, 62, and references therein]. Some works have studied tree-based shape functions [33] and neural basis functions [1, 58].

**GAMs with all pairwise interactions.** In this thread, [9] study low-rank decomposition with neural network shape functions; [46] fit all pairwise interactions using shared neural bases.

**Sparse GAMs with interactions.** [31] introduced COSSO, which penalizes the sum of the Sobolev norms of the functional components, producing sparse models. [23] propose ELAAN, which is an $\ell_0$-regularized formulation with smooth cubic splines. [23] demonstrate the usefulness of their approach in a regression setting in terms of compact component selection and efficiency on a large-scale Census survey response prediction. [34, 40, 6] explore tree-based shape functions for fitting additive models with sparse interactions. [34] consider *all* main effects and a subset of pairwise interactions; the subset of interactions are selected via greedy stage-wise interaction detection heuristics. [40] provide an efficient implementation of the above approach as Explainable Boosting Machines (EBMs). [6] propose NODE-GA$^2$M: an end-to-end learning approach with differentiable neural oblivious decision (NODE) trees [44]. Component selection in NODE-GAM is achieved by constraining the number of trees, and each tree learns to use one or two features via entmax transformation [43].

**Structural Constraints.** Structural interpretability constraints such as hierarchical interactions, have been studied for both linear settings [3, 30, 59, 17] and nonparametric settings [45, 60, 10, 23]. We briefly review prior work on nonparametric hierarchical interactions as it relates to this paper. [60] proposed GAMI-Net, which is a multi-stage neural-based approach that fits *all* main effects and a subset of Top-*k* interaction effects, selected via a fast interaction screening method [34]. Amongst this screened set, interactions that satisfy the weak hierarchy are later used to fit interaction effects. They also prune some main and interaction effects after training based on a variation-based ranking measure. [10] proposed SIAN, which uses Archipelago [54] to measure the strength of each main and interaction effect from a *trained* DNN, and then screens (i.e., selects a subset of candidate main and interaction effects) using Archipelago scores to identify main effects and interaction effects that obey strong hierarchy. Then, it fits a GAM model with the screened main and interaction effects. [60] and [10] only support screening of interactions obeying hierarchy *before* the training for interaction effects is done. None of these approaches impose hierarchy *while* training with interactions. [23] with their ELAAN framework also consider strong hierarchy in the presence of $\ell_0$-regularized formulation with splines for the least squares loss (regression). ELAAN has a two-stage approach: It selects a candidate set of interactions and then applies commercial mixed integer programming solvers to learn sparse interactions under a hierarchy constraint. This approach would require customized algorithms to adapt to different loss objectives e.g., multiclass classification. To our knowledge, current techniques for sparse hierarchical (nonparametric) interactions are not based on end-to-end differentiable training: they can be limited in flexibility and scalability—a gap we intend to fill in this work.

Table 1: Relevant work on sparse GAMs with Interactions. Models in rows 1-2 have some variable selection but no hierarchy; models in row 3-5 have screening-based approaches for hierarchy.

| Paper | Selection Method | | Structural Constraints | | Reg. | Classif. | | Shape Functions | Statistical Properties | Scalable |
|---|---|---|---|---|---|---|---|---|---|---|
| | Main | Interactions | Weak Hier. | Strong Hier. | | Bin. | Multi | | | |
| EBM [40] | None | Greedy | ✗ | ✗ | ✓ | ✓ | ✓ | trees | ✗ | ✓ |
| NODE-GA$^2$M [6] | Entmax+Anneal | | ✗ | ✗ | ✓ | ✓ | ✓ | trees | ✗ | ✓ |
| GAMI-Net [60] | Prune | Screening | Screening | ✗ | ✓ | ✓ | ✗ | neural | ✗ | ✗ |
| SIAN [10] | None | Screening | ✗ | Screening | ✓ | ✓ | ✗ | neural | ✓ | ✗ |
| ELAAN [23] | Group L0 | | ✗ | Screening+Convex Relax. | ✓ | ✗ | ✗ | splines | ✓ | ✓ |
| `GRAND-SLAMIN` | Binary Variables | | End-to-end | End-to-end | ✓ | ✓ | ✓ | trees | ✓ | ✓ |

Hier.=Hierarchy, Reg.=Regression, Classif.=Classification, Bin.=Binary, Relax.=Relaxation

Note [9] and [10] also consider higher-order interactions (beyond two-way ones), which can be hard to interpret. For convenience, Table 1 summarizes some relevant work on Sparse GAMs with interactions and possible structural constraints.

## 3   Problem Formulation

We first present in Sec. 3.1 an alternative formulation of GAMs with interactions using binary variables for imposing sparsity and structural constraints. Next, in Sec. 3.2, we present a smooth reformulation of the objective that can be solved with first-order gradient-based methods.

### 3.1   An optimization formulation with binary variables

We first present an alternative formulation of GAMs with interactions under sparsity with/without additional structured hierarchy constraints. Let us consider the parameterization:

$$f = \sum_{j \in [p]} f_j(x_j) z_j + \sum_{(j,k) \in \mathcal{I}} f_{j,k}(x_j, x_k) q(z_j, z_k, z_{j,k}), \tag{4}$$

with main effects $f_j(\cdot)$, interaction effects $f_{j,k}(\cdot)$ and binary gates $z_j$ and $q(z_j, z_k, z_{j,k})$. We consider three different parameterizations for $q(\cdot)$, satisfying the following different constraints:

$$\text{No structural constraint:} \quad q(z_j, z_k, z_{j,k}) \stackrel{\text{def}}{=} z_{j,k}, \tag{5}$$

$$\text{Weak hierarchy:} \quad q(z_j, z_k, z_{j,k}) \stackrel{\text{def}}{=} (z_j + z_j - z_j z_k) z_{j,k}, \tag{6}$$

$$\text{Strong hierarchy:} \quad q(z_j, z_k, z_{j,k}) \stackrel{\text{def}}{=} z_j z_k z_{j,k}. \tag{7}$$

The binary gates $z_j$ and $q(z_j, z_k, z_{j,k})$ play the role of selection. In particular, when $z_j = 0$, the corresponding $j$-th main effect $f_j(\cdot)$ is excluded from our additive model (4). Similarly, when $q(z_j, z_k, z_{j,k}) = 0$, the corresponding $(j, k)$-th interaction effect $f_{j,k}(\cdot)$ is excluded. Then, we can formulate the regularized objective as:

$$\min_{\substack{\{f_j\}, \{f_{j,k}\}, \\ \{z_j\} \in \{0,1\}^p, \{z_{j,k}\} \in \{0,1\}^{|\mathcal{I}|}}} \hat{\mathbb{E}}[\ell(y, f)] + \lambda \Big( \sum_{j \in [p]} z_j + \alpha \sum_{(j,k) \in \mathcal{I}} z_{j,k} \Big), \tag{8}$$

where the first term denotes empirical loss over the training data, the penalty term controls model sparsity: $\lambda \geq 0$ is the selection penalty, $\alpha \in [1, \infty)$ controls the relative selection strength of main and interaction effects. We refer to the framework in (8) under the different constraints (5)-(7) as GRAND-SLAMIN[1]. We discuss extension of this framework to third-order interactions in Supplement Sec. D. However, we do not consider third-order interactions in our experiments as third-order interactions are hard to interpret.

The formulation in (8) with binary variables $z_j$, $z_{j,k}$ and functions $f_j$, $f_{j,k}$ with any of the constraint sets (5)-(7) is a challenging discrete optimization problem (with binary variables) and is not amenable to differentiable training via SGD (for example). Sec. 3.2 explores approximate solutions to (8) using a smooth reformulation of the binary variables. Intuitively, we rely on continuous relaxations of the binary variables $z$'s and parameterize $f$'s with smooth tree-based shape functions. The reformulation allows us to use first-order methods.

### 3.2   A Smooth Reformulation of Problem (8)

We discuss a smooth reformulation of the objective in (8). We describe an approach to parameterize the continuous relaxation of the binary variables $z_j$, $z_{j,k}$ with a Smooth-Step function [18] and use smooth tree-based shape functions to model $\{f_i\}$, $\{f_{j,k}\}$.

#### 3.2.1   Relaxing Binary Variables with Smooth Gates

We present an approach to smooth the binary gates $z$'s in (8) using a smooth-step function [18], which we define next.

---

[1]GRAND-SLAMIN stands for GeNeRAlizeD Sparse Learning of Additive Models with INteractions.

**Smooth-Step Function.** Smooth-step function is a continuously differentiable function, similar in shape to the logistic function. However, unlike the logistic function, the smooth-step function can output 0 and 1 exactly for sufficiently large magnitudes of the input (see Appendix B for details). This function has been used for smoothing binary representations for conditional computation [18, 19, 21].

We parameterize each of the $z$'s in (8) as $S(\mu)$, where $\mu \in \mathbb{R}$ is a learnable parameter and $S(\cdot)$ denotes the Smooth-step function. We parameterize the additive function as: $f = \sum_{j \in [p]} f_j(x_j) S(\mu_j) + \sum_{(j,k) \in \mathcal{I}} f_{j,k}(x_j, x_k) q(S(\mu_j), S(\mu_k), S(\mu_{j,k}))$ and optimize the following objective:

$$\min_{\substack{\{f_j\}, \{f_{j,k}\}, \\ \{\mu_j\} \in \mathbb{R}^p, \{\mu_{j,k}\} \in \mathbb{R}^{|\mathcal{I}|}}} \hat{\mathbb{E}}[\ell(y, f)] + \lambda (\sum_{j \in [p]} S(\mu_j) + \alpha \sum_{(j,k) \in \mathcal{I}} S(\mu_{j,k})). \tag{9}$$

Note that $S(\mu_j)$ and $S(\mu_{j,k})$ are continuously differentiable, so the formulation in (9) is amenable to first-order gradient-based methods (e.g, SGD).

**Achieving binary gates.** To encourage each of the $S(\mu_j)$'s and $S(\mu_{j,k})$'s to achieve binary state (and, not fractional) by the end of training, we add an entropy regularizer $\tau(\sum_{j \in [p]} \Omega(S(\mu_j)) + \sum_{(j,k) \in \mathcal{I}} \Omega(S(\mu_{j,k})))$ where $\Omega(S(\mu)) = -(S(\mu) \log S(\mu) + (1 - S(\mu)) \log(1 - S(\mu)))$ and $\tau \geq 0$ controls how quickly each of the gates $S(\mu)$ converges to a binary $z$.

### 3.2.2 Soft trees

We use soft trees [25, 24, 53, 11]—based on hyperplane splits (univariate or bivariate) and constant leaf nodes — as shape functions to parameterize the main effects $f_j(\cdot)$ and the pairwise interaction effects $f_{j,k}(\cdot)$. See Figure 1 for an illustration. Soft trees were introduced as hierarchical mixture of experts by [25] and further developed by [24, 53, 11]. They allow for end-to-end learning [27, 18, 22]. They also have efficient implementations when learning tree ensembles [22]. A detailed definition of soft tress is given in Appendix A.

## 4 Efficient Implementation

We discuss a fast implementation of our approach. The key elements are: (i) Tensor parameterization of trees, (ii) Sparse backpropagation, and (iii) Complementary screening heuristics.

**Tensor Parameterization of Additive Effects.** Typically, in neural-based additive model implementations [1, 60], a separate network module is constructed for each shape function. The outputs from each model are sequentially computed and combined additively. This can create bottleneck in scaling these models. Some recent approaches e.g., [10] try to work around this approach

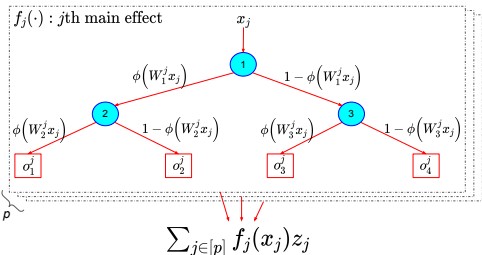

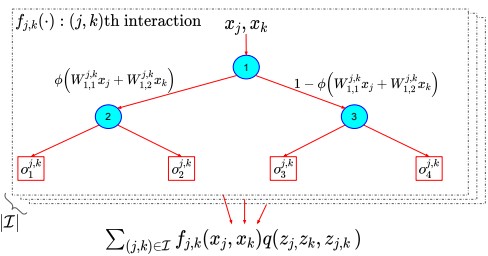

Figure 1: Modeling main and interaction effects with soft trees. $\phi$ denotes the sigmoid activation function. We omit biases in split nodes for brevity. For interaction effect, $W_{i,1}^{j,k} \in \mathbb{R}$ and $W_{i,2}^{j,k} \in \mathbb{R}$ denote the weights in $i$-th node of the $(j, k)$-th tree.

by constructing a large block-wise network to compute representations of all shape functions simultaneously. However, this comes at a cost of large memory footprint. For tree-based shape functions, drawing inspiration from [22], we can implement a tensor-based formulation of all shape functions, leveraging the fact that each tree has the same depth. This parameterization can exploit GPU-friendly parallelization in computing all shape functions simultaneously without increasing memory footprints.

**Sparse Backpropagation.** We use first-order optimization methods (e.g., SGD and its variants) to optimize GRAND-SLAMIN. Typically, a main computational bottleneck in optimizing GAMs with all pairwise interactions via standard gradient-based backpropagation methods is the computation

of forward pass and gradient computations with respect to all additive components (both main and interaction effects). This can hinder training large GAMs with interactions. We exploit the sparsity in GRAND-SLAMIN via the sparsity in the smooth-step function and its gradient during training.

Recall that $S(\mu_j)$'s and $S(\mu_{jk})$'s play a role of selection in a smoothed fashion. In the early stages of training, $S(\mu_j)$'s and $S(\mu_{jk})$'s are all in the range $(0, 1)$. As the optimization proceeds, due to the entropic regularization and selection regularization, $S(\mu_j)$'s and $S(\mu_{jk})$'s progressively achieve binary state $\{0, 1\}$ — the gradient with respect to $\mu_j$ and $\mu_{jk}$ also reaches 0 because of the nature of smooth-step function $S(\cdot)$. All the additive components corresponding to the selection variables that reached 0 can be removed from both the forward and the backward computational graph. This sparse backpropagation approach can provide large speedups on training times up to a factor of $10\times$ over standard backpropagation. Additionally, there is progressively a reduced memory footprint during the course of training in comparison to standard backpropagation.

The approach outlined above (use of Smooth-Step function for selection) when specialized to additive models allows us to implement GPU-friendly sparse backpropagation — this makes our work different from [18], which does not support GPU training.

**Screening.** We describe how screening approaches prior to training, can complement our sparse backpropagation approach when the number of all pairwise interactions is large e.g., of the order $100,000$. Fast screening methods based on shallow-tree like models are proposed in [34] for identifying prominent pairwise interactions. These are used by various toolkits e.g., EBM [40], GAMI-Net [60]. These screening approaches are complementary to our selection approach with indicator variables. We used CART [4] for each pairwise interaction and sorted the interaction effects based on AUC performance to select an initial screened set of interaction effects. In particular, we can consider $\mathcal{I}$ to be a screened subset (e.g., $10,000$) of all pairwise interaction effects of the order $100,000$. Then, we run our end-to-end learning framework under the component selection constraints on the main and screened interaction effects. We observe that such screening can be beneficial for multiple reasons: (i) The training time can be reduced further by $3\times - 5\times$. (ii) The memory footprint of the model reduces by $10\times$. (iii) There is no loss in accuracy—the accuracy can sometimes improve with screening — see ablation study in Supplement Sec. F.6. Note that even with screening, our approach is different from GAMI-Net as they directly screen to a much smaller set of interactions e.g., $500$ and there is no component selection when training with these interactions.

## 5 Statistical Theory

In this section, we explore the statistical properties of our additive model with soft tree shape functions. We assume that observations are noisy versions of some (unknown) underlying noiseless data. We do not assume the noiseless data comes from a tree-based model. We show that if our model can approximate the true noiseless data well, the prediction error resulting from the noise converges to zero as the number of observations grows. In particular, for $n$ data observations $(\boldsymbol{x}_i, y_i)_{i=1}^n$ we consider a sparsity-constrained version of Problem (8) with the least-squares loss as given by

$$f_{(t)}(\boldsymbol{y}) \in \text{argmin} \quad \sum_{i=1}^n [y_i - \sum_{j \in [p]} f_j(x_{i,j})z_j - (t-1) \sum_{(j,k) \in \mathcal{I}} f_{j,k}(x_{i,j}, x_{i,k})q(z_j, z_k, z_{j,k})]^2, \quad (10)$$

$$\text{s.t.} \quad \{z_j\} \in \{0,1\}^p, \{z_{j,k}\} \in \{0,1\}^{|\mathcal{I}|}, \quad \sum_{j \in [p]} z_j + \sum_{(j,k) \in \mathcal{I}} q(z_j, z_k, z_{j,k}) \leq s,$$

for $t \in \{1, 2\}$, where $f_j, f_{j,k}$ are depth-$d$ soft trees (cf. Section 3.2.2). For $t = 1$, Problem (10) simplifies to a main effects model, while $t = 2$ corresponds to the model with pairwise interactions. We study the case with no hierarchy constraints here, $q(z_j, z_k, z_{j,k}) = z_{j,k}$ as in (5). We expect our approach to extend to more general cases, but we do not pursue that here.

**Model Setup.** We assume for $i \in [n]$, the data is bounded, $\|\boldsymbol{x}_i\|_2 \leq 1$. The noisy observations are given as $y_i = h^*(\boldsymbol{x}_i) + \varepsilon_i = y_i^* + \varepsilon_i$ where $h^*$ is the unknown underlying generative model (need not be a tree ensemble), and $\varepsilon_i \overset{\text{iid}}{\sim} \text{Normal}(0, \sigma^2)$ are noise values. Suppose $f$ is a feasible solution for Problem (10). Let $\mathcal{U}, \mathcal{L}$ denote the set of internal nodes and leaves for a depth $d$ tree, and let $(\boldsymbol{W}^j, \boldsymbol{o}^j)$ be the set of weights corresponding to internal nodes and leaves of tree $j$ i.e. $f_j(x_j)$ in this solution, where $\boldsymbol{W}^j \in \mathbb{R}^{|\mathcal{U}|}, \boldsymbol{o}^j \in \mathbb{R}^{|\mathcal{L}|}$. We also let $W_i^j, o_l^j$ be the weights corresponding to node $i \in \mathcal{U}$ and leaf $l \in \mathcal{L}$ in tree $j$, respectively. We define $\boldsymbol{W}_i^{j,k}, o_l^{j,k}$ for $f_{j,k}(x_j, x_k)$ and $i \in \mathcal{U}, l \in \mathcal{L}$

in a similar fashion, where $\boldsymbol{W}^{j,k} \in \mathbb{R}^{|\mathcal{U}| \times 2}, \boldsymbol{o}^{j,k} \in \mathbb{R}^{|\mathcal{L}|}$. See Figure 1 for an illustration. Define

$$\bar{u}(f) = \max_{\substack{j \in [p], i \in \mathcal{U} \\ z_j = 1}} |W_i^j| \vee \max_{\substack{j \in [p], l \in \mathcal{L} \\ z_j = 1}} |o_l^j| \vee \max_{\substack{j,k \in [p], i \in \mathcal{U} \\ z_{j,k} = 1}} \|\boldsymbol{W}_i^{j,k}\|_2 \vee \max_{\substack{j,k \in [p], l \in \mathcal{L} \\ z_{j,k} = 1}} |o_l^{j,k}|$$

where $\vee$ denotes maximum. Conceptually, $\bar{u}(f)$ is the largest weight (in absolute value) that appears in all main effect and interaction soft trees. Let $\hat{f} = f_{(t)}(\boldsymbol{y})$ be the estimator resulting from the noisy data, and $f^* = f_{(t)}(\boldsymbol{y}^*)$ be the oracle estimator that is the best approximation to the noiseless data among feasible solutions of (10) (i.e., among sparse additive tree models). We assume the following:

**(A1)** The activation function $\phi : \mathbb{R} \mapsto [0,1]$ for soft trees is $L$-Lipschitz for some $L > 0$.
**(A2)** There exists $B > 0$ such that $\bar{u}(\hat{f}) \vee \bar{u}(f^*) \leq B$.

Assumption **(A2)** ensures the trees resulting from the data are uniformly bounded. This is a mild assumption as in practice, the data is bounded and the resulting trees are also bounded as a result.

**Main Results.** Our first result is for a general setup, where $p$ can be much larger than $n$.

**Theorem 1.** *Let $\hat{f}, f^*$ be as defined above and take $A = 2 \vee 2^{d+2}B \vee B^2 dL2^{d+3}$ and $a = \exp(-1)$. Under the model setup, assume $s \geq 1$ and $\sigma \gtrsim 1$.[2] Then,*
*(1) For $t = 1$, if $n \gtrsim \sigma^2(s \log p + s^2 A^2)$, then with high probability*

$$\frac{1}{n} \sum_{i=1}^n (\hat{f}(\boldsymbol{x}_i) - y_i^*)^2 \lesssim \frac{1}{n} \sum_{i=1}^n (f^*(\boldsymbol{x}_i) - y_i^*)^2 + \frac{\sigma^{4/3}}{n^{2/3}} \left( (s \log p)^{2/3} + (sA)^{4/3} \right).$$

*(2) For $t = 2$, if $n \gtrsim \sigma^{2+2a}(s \log p + s^3 A^3)$, then with high probability*

$$\frac{1}{n} \sum_{i=1}^n (\hat{f}(\boldsymbol{x}_i) - y_i^*)^2 \lesssim \frac{1}{n} \sum_{i=1}^n (f^*(\boldsymbol{x}_i) - y_i^*)^2 + \frac{\sigma^{2(1+a)/(2+a)}}{n^{1/(2+a)}} \left( \sqrt{s \log p} + (sA)^{3/2} \right)^{2/(2+a)}.$$

Theorem 1 presents non-asymptotic bounds for the performance of our method. Particularly, this theorem shows that under a well-specified model where $y_i^* = f^*(\boldsymbol{x}_i)$, prediction error rates of $n^{-2/3}$ and $n^{-1/(2+a)} \approx n^{-0.42}$ are achievable for main effects and interaction models, respectively. Particularly, this implies that the prediction error of our estimator (resulting from the noise in observations) converges to zero as we increase the total number of samples, $n$. We note that although the rate from the main effects model is sharper, such models might be too simple to capture the noiseless data, leading to larger oracle error. Therefore, in practice, the interactions model can lead to better performance. Next, we show that under the asymptotic assumption that $n \to \infty$, the error rate for the interaction models can be further improved.

**Theorem 2.** *Under the model setup with $t = 2$, assume all parameters are fixed except $n$. For any positive sequence $a_n$ with $\lim_{n\to\infty} a_n = 0$, there exists a positive sequence $b_n$ with $\lim_{n\to\infty} b_n = 0$ such that if $n \gtrsim b_n^{-1/2(2+a_n)}$ then with high probability*

$$\frac{1}{n} \sum_{i=1}^n (\hat{f}(\boldsymbol{x}_i) - y_i)^2 \lesssim \frac{1}{n} \sum_{i=1}^n (f^*(\boldsymbol{x}_i) - y_i^*)^2 + \frac{1}{n^{1/(2+a_n)}}.$$

Theorem 2 shows that essentially when $n \to \infty$ and other parameters in the problem stay constant, an error rate of $n^{-0.5}$ is achievable for the interactions model which improves upon Theorem 1.

**Discussion of previous work:** As stated, our results are the first to present prediction bounds specialized to learning sparse additive models with soft trees. Moreover, as we discuss in Appendix C.5, our upper bounds (rates) generally align with the best known rates available in the literature for learning main effects and interactions, such as [23, 52], or hold under milder assumptions.

## 6   Experiments

We study the performance of GRAND-SLAMIN on 16 real-world datasets and compare against relevant baselines for different cases. We make the following comparisons:

1. Performance comparison of GRAND-SLAMIN without/with structural constraints against existing toolkits for sparse GAMs with interactions.

---

[2]The notation $\lesssim, \gtrsim$ depict an inequality holds up to a universal constant independent of problem data.

(a) Toolkits that support sparse interactions: EB$^2$M [40] and NODE-GA$^2$M [6]
(b) Toolkits that support hierarchy constraints: GAMI-Net [60] and SIAN [10]
2. Variable selection comparison against the competing toolkits.
3. Computational scalability of `GRAND-SLAMIN` toolkit with sparse backpropagation.
4. Variance reduction with structural constraints

Additional results are included in Supplement Sec. F that study (i) comparison with full complexity models in F.1, (ii) comparison with GAMs with all pairwise interactions in F.2, (iii) comparison with Group Lasso selection approach in F.3, (iv) choice of shape functions in F.4, (v) effect of entropy on performance and component selection in F.5, and (vi) effect of screening on training times, memory and performance in F.6.

**Datasets.** We use a collection of 16 open-source classification datasets (8 binary, 6 multiclass and 2 regression) from various domains. We consider datasets with a wide range of number of all pairwise interactions $10 - 200000$. A summary of the datasets is in Table E.1 in the Appendix.

**Tuning Details.** For all the experiments, we tune the hyperparameters using Optuna [2] with random search on a held-out validation set. We compute statistical averages across multiple runs for the optimal hyperparameters for all models. In particular, we report median test ROC AUC across 10 runs along with the mean absolute deviation (MAD). Additional details are in the Appendix E.

Table 2: Test ROC AUC of `GRAND-SLAMIN`, EB$^2$M and NODE-GA$^2$M. We report median along with mean absolute deviation across 10 runs.

| Dataset | EB$^2$M | NODE-GA$^2$M | GRAND-SLAMIN |
|---|---|---|---|
| Magic | $93.12 \pm 0.001$ | $\mathbf{94.27} \pm 0.13$ | $93.86 \pm 0.30$ |
| Adult | $91.41 \pm 0.0004$ | $\mathbf{91.75} \pm 0.14$ | $91.54 \pm 0.14$ |
| Churn | $91.97 \pm 0.005$ | $89.62 \pm 5.61$ | $\mathbf{92.40} \pm 0.41$ (SH) |
| Satimage | $97.65 \pm 0.0007$ | $98.70 \pm 0.07$ | $\mathbf{98.81} \pm 0.04$ |
| Texture | $99.81 \pm 0.0004$ | $\mathbf{100.00} \pm 0.00$ | $\mathbf{100.00} \pm 0.00$ |
| MiniBooNE | $97.86 \pm 0.0001$ | $\mathbf{98.44} \pm 0.02$ | $97.77 \pm 0.05$ (WH) |
| Covertype | $90.08 \pm 0.0003$ | $95.39 \pm 0.12$ | $\mathbf{98.11} \pm 0.08$ |
| Spambase | $\mathbf{98.84} \pm 0.01$ | $98.78 \pm 0.06$ | $98.55 \pm 0.07$ (SH) |
| News | $73.03 \pm 0.002$ | $\mathbf{73.53} \pm 0.06$ | $73.24 \pm 0.04$ (SH) |
| Optdigits | $99.79 \pm 0.0003$ | $99.93 \pm 0.02$ | $\mathbf{99.98} \pm 0.0$ |
| Bankruptcy | $\mathbf{93.85} \pm 0.01$ | $92.02 \pm 1.03$ | $92.51 \pm 0.54$ (WH) |
| Madelon | $88.04 \pm 0.02$ | $60.07 \pm 0.82$ | $\mathbf{89.25} \pm 1.03$ (WH) |
| Activity | $74.96 \pm 8.77$ | $\mathbf{99.86} \pm 0.04$ | $99.24 \pm 1.45$ |
| Multiple | $\mathbf{99.96} \pm 0.0002$ | $99.94 \pm 0.02$ | $99.95 \pm 0.02$ |

## 6.1 Prediction Performance

**Comparison with EB$^2$M and NODE-GA$^2$M.** We first study the performance of our model in comparison to two tree-based state-of-the-art toolkits which support sparse GAMs with interactions without any structural constraints e.g., EB$^2$M and NODE-GA$^2$M. We report the ROC AUC performance in Table 2. Our model outperforms EB$^2$M in 10 out of 14 datasets. Our model is also competitive with NODE-GA$^2$M as it can outperform in $50\%$ of the datasets. In summary, our results in Table 2 show that we are at par with state-of-the-art methods for unstructured component selection.

Table 3: Test ROC AUC for `GRAND-SLAMIN` with structural constraints i.e., (6) or (7), GAMI-Net and SIAN. We report median across 10 runs along with mean absolute deviation.

| Dataset\Model | GAMI-Net | SIAN | GRAND-SLAMIN | |
|---|---|---|---|---|
| | WH | SH | WH | SH |
| Magic | $91.72 \pm 0.05$ | $93.02 \pm 0.06$ | $93.16 \pm 0.55$ | $\mathbf{93.37} \pm 0.16$ |
| Adult | $91.01 \pm 0.04$ | $90.67 \pm 0.05$ | $91.34 \pm 0.32$ | $\mathbf{91.46} \pm 0.15$ |
| Churn | $90.05 \pm 0.77$ | $\mathbf{92.98} \pm 0.20$ | $92.28 \pm 0.75$ | $92.40 \pm 0.41$ |
| Spambase | $\mathbf{98.67} \pm 0.04$ | $98.28 \pm 0.04$ | $98.45 \pm 0.15$ | $98.55 \pm 0.07$ |
| MiniBooNE | $96.11 \pm 0.41$ | $95.90$ | $\mathbf{97.77} \pm 0.05$ | $97.62 \pm 0.30$ |
| News | $72.54 \pm 0.05$ | $72.28$ | $73.15 \pm 0.08$ | $\mathbf{73.24} \pm 0.04$ |
| Bankruptcy | $92.46 \pm 0.12$ | $90.71$ | $\mathbf{92.51} \pm 0.54$ | $90.45 \pm 1.87$ |
| Madelon | $88.14 \pm 0.94$ | $83.18$ | $\mathbf{89.25} \pm 1.03$ | $86.23 \pm 1.89$ |

WH=Weak Hierarchy, SH=Strong Hierarchy.
For SIAN, for some of the larger datasets (row 5-9),
we use the number for best trial as SIAN takes
$\sim 24$ hours on V100 Tesla GPU.

Our key advantage is to do hierarchical interactions, which NODE-GA$^2$M and EB$^2$M can not support. Additionally, we can achieve faster training times (Sec. 6.3) and improve on variable selection (Sec. 6.2) than NODE-GA$^2$M and EB$^2$M.

**Structural constraints: Weak and Strong Hierarchy.** Next, we study our method with structural constraints i.e., (6) for weak hierarchy or (7) for strong hierarchy. We compare against two competing neural-based state-of-the-art methods for sparse GAMs with hierarchical interactions: (i) GAMI-Net with support for weak hierarchy, and (ii) SIAN with support for strong hierarchy. We omit multiclass datasets as both GAMI-Net and SIAN do not support them. We report the ROC AUC performance in Table 3. Our models outperform GAMI-Net and SIAN in 7/8 datasets.

Additionally, our models are much more compact in terms of overall number of parameters — our tree-based shape functions have $100\times$ and $10\times$ smaller number of parameters than the neural-based

shape functions used by GAMI-Net and SIAN respectively. Moreover, our toolkit is significantly faster than SIAN and GAMI-Net on larger datasets — see Sec. 6.3.

Additionally, we compare interpretable modeling toolkits with full complexity models e.g., deep neural network (DNN), in Supplement Sec. F.1. We observed interpretable models to outperform full complexity models on these datasets. We also compare our toolkit that fits sparse components with toolkits that fit all pairwise interactions e.g., NA$^2$M, NB$^2$M and SPAM in Supplement Sec. F.2. GRAND-SLAMIN generally outperform these methods with enhanced interpretability due to sparsity and structural constraints. We also study how our toolkit performs when we replace soft tree shape functions with MLP shape functions in Supplement Sec. F.4. Interestingly, we observe that soft trees seem to have an edge when the parameters are matched with MLP.

## 6.2 Variable Selection

We evaluate the performance of our models in terms of feature selection. We report the number of features selected in Table 4 by each toolkit for sparse GAMs with interactions. We see that GRAND-SLAMIN with structural constraints, in particular strong hierarchy, can significantly reduce the number of features selected by the GAMs with interactions model. For example, on Bankruptcy datasets, GRAND-SLAMIN achieves feature compression up to a factor of $8\times$ over state-of-the-art GAM toolkits. Having fewer features reinforces the usefulness of additive models as being interpretable.

Table 4: Number of features used by GRAND-SLAMIN without/with additional structural constraints and competing approaches. Hyphen (-) indicates multiclass classification is not supported by GAMI-Net and SIAN.

| Dataset\Model | EB$^2$M | NODE GA$^2$M | GAMI Net | SIAN | GRAND-SLAMIN None | WH | SH |
|---|---|---|---|---|---|---|---|
| Magic | $10 \pm 0$ | $10 \pm 0$ | $10 \pm 0$ | $10 \pm 0$ | $10 \pm 0$ | $9 \pm 1$ | $\mathbf{7 \pm 0}$ |
| Adult | $14 \pm 0$ | $14 \pm 0$ | $14 \pm 1$ | $14 \pm 0$ | $13 \pm 1$ | $\mathbf{11 \pm 1}$ | $\mathbf{11 \pm 1}$ |
| Churn | $19 \pm 0$ | $19 \pm 0$ | $18 \pm 2$ | $19 \pm 0$ | $19 \pm 0$ | $\mathbf{11 \pm 1}$ | $12 \pm 2$ |
| Satimage | $36 \pm 0$ | $36 \pm 0$ | – | – | $36 \pm 0$ | $36 \pm 0$ | $\mathbf{22 \pm 2}$ |
| Texture | $40 \pm 0$ | $40 \pm 0$ | – | – | $40 \pm 0$ | $37 \pm 2$ | $\mathbf{17 \pm 2}$ |
| MiniBooNE | $50 \pm 0$ | $50 \pm 0$ | $\mathbf{16 \pm 12}$ | $34$ | $50 \pm 0$ | $50 \pm 0$ | $28 \pm 3$ |
| Covertype | $54 \pm 0$ | $54 \pm 0$ | – | – | $\mathbf{34 \pm 1}$ | $54 \pm 1$ | $54 \pm 0$ |
| Spambase | $57 \pm 0$ | $57 \pm 0$ | $\mathbf{52 \pm 2}$ | $55 \pm 1$ | $57 \pm 0$ | $56 \pm 3$ | $54 \pm 2$ |
| News | $58 \pm 0$ | $58 \pm 0$ | $\mathbf{47 \pm 1}$ | $52$ | $58 \pm 0$ | $58 \pm 0$ | $58 \pm 0$ |
| Optdigits | $64 \pm 0$ | $64 \pm 0$ | – | – | $64 \pm 0$ | $64 \pm 0$ | $\mathbf{59 \pm 1}$ |
| Bankruptcy | $95 \pm 0$ | $95 \pm 0$ | $60 \pm 15$ | $69$ | $95 \pm 0$ | $60 \pm 26$ | $\mathbf{7 \pm 16}$ |
| Madelon | $500 \pm 0$ | $500 \pm 0$ | $61 \pm 56$ | $490$ | $26 \pm 19$ | $\mathbf{19 \pm 15}$ | $24 \pm 9$ |
| Activity | $533 \pm 0$ | $346 \pm 6$ | – | – | $182 \pm 15$ | $440 \pm 22$ | $\mathbf{159 \pm 21}$ |
| Multiple | $649 \pm 0$ | $649 \pm 0$ | – | – | $648 \pm 1$ | $\mathbf{629 \pm 9}$ | $649 \pm 0$ |

WH=Weak Hierarchy, SH=Strong Hierarchy.

## 6.3 Computational Scalability

Next, we discuss the scalability of GRAND-SLAMIN.

**Sparse backpropagation.** We highlight the usefulness of our efficient approach with sparse backpropagation in Figure 2 on Activity dataset. We show in Figure 2[a] that during the course of training, the number of selected components (selected via binary variables $z$'s and $q(\cdot)$'s) becomes progressively smaller. This leads to much faster computations at each epoch in Figure 2[b] due to

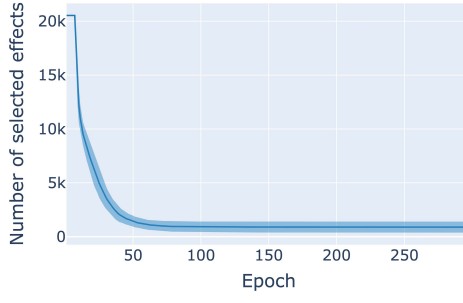

(a) Number of selected effects at each epoch.

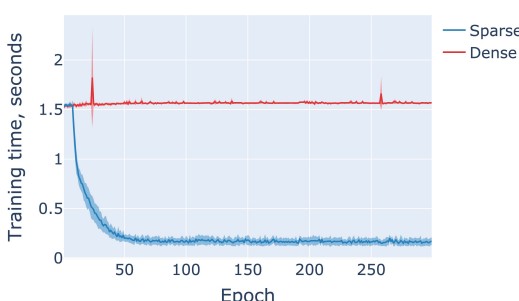

(b) Training time (seconds) for each epoch.

Figure 2: GRAND-SLAMIN with standard (dense) backpropagation vs sparse backpropagation on Activity dataset. (a) shows the number of nonzero effects: $\sum_j z_j + \sum_{(j,k) \in \mathcal{I}} q(z_j, z_k, z_{j,k})$ and (b) shows the time for each epoch during the course of training.

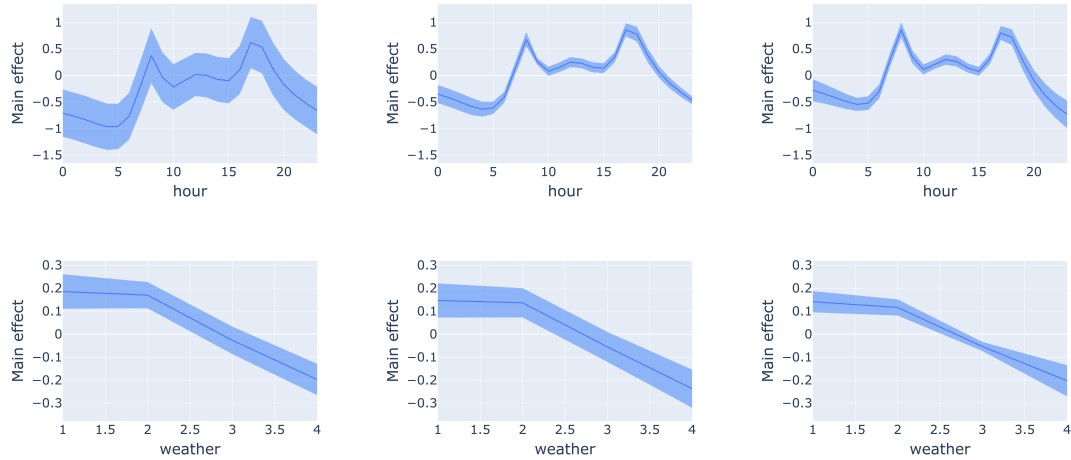

Figure 3: Estimated main effects in the presence of interaction effects on bikesharing dataset [Left] without hierarchy, [Middle] weak hierarchy and [Right] strong hierarchy. Strong hierarchy has the smallest error bars.

sparse forward and backward passes. By exploiting sparsity during training, we can get $10\times$ faster training times than with standard backpropagation.

**Comparison with other toolkits.** Our toolkit is highly competitive in terms of training times with all existing tree-based and neural-based toolkits for sparse GAMs with interactions. For example, on Madelon dataset, we are $15\times$ faster than NODE-GA$^2$M, $20\times$ faster than EBM, $1300\times$ faster than SIAN and $25\times$ faster than GAMI-Net. See Supplement Sec. F.7 for more detailed timing comparisons across multiple datasets. Note that, in addition, we can also handle the case of structured interactions — extending the flexibility of existing end-to-end training methods.

### 6.4 Variance Reduction with Structural Constraints

We provide a visualization study to further highlight an important contribution of our work. In particular, our framework can support models with structural constraints. Hence, we study the effect of these constraints on the stability of learning main effects (in the presence of interactions) when these structural constraints are imposed or not. For this exercise, we consider bikesharing dataset. We visualize some of the main effects in the presence/absence of hierarchy in Figure 3. Note that for visualization, we used the purification strategy [29, 6] post-training that pushes interaction effects into main effects if possible. We can observe in Figure 3 that when additional hierarchy constraints are imposed, the error bars are much more compact across different runs. This can potentially increase the trust you can have on the model for deriving interpretability insights. We show additional visualizations on another dataset (American Community Survey from US Census Planning Database 2022 [55]) to show the same behavior in Supplement Section G.

## 7 Conclusion

We introduce GRAND-SLAMIN: a novel and flexible framework for learning sparse GAMs with interactions with additional structural constraints e.g., hierarchy. This is the first approach to do end-to-end training of nonparameteric additive models with hierarchically structured sparse interactions. Our formulation uses binary variables to encode combinatorial constraints. For computational reasons, we employ smoothing of the indicator variables for end-to-end optimization with first-order methods (e.g., SGD). We propose sparse backpropagation, which exploits sparsity in the nature of the smoothing function in a GPU-compatible manner and results in $10\times$ speedups over standard backpropagation. We present non-asymptotic prediction bounds for our estimators with tree-based shape functions. Numerical experiments on a collection of 16 real-world datasets demonstrate the effectiveness of our toolkit in terms of prediction performance, variable selection and scalability.

## Acknowledgments and Disclosure of Funding

This research was supported in part, by grants from the Office of Naval Research (N000142112841), and Liberty Mutual Insurance. The authors acknowledge the MIT SuperCloud [48] for providing HPC resources that have contributed to the research reported within this paper.

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

# Supplementary Material for *GRAND-SLAMIN' Interpretable Additive Modeling with Structural Constraints*

## A    Definition of Soft Trees

Formally, an interaction soft tree with the set of (internal) nodes $\mathcal{U}$ and the set of leaves $\mathcal{L}$ is a function such as $f(\cdot; \boldsymbol{W}, \boldsymbol{o}) : \mathbb{R}^2 \mapsto \mathbb{R}$ where $\boldsymbol{W} \in \mathbb{R}^{|\mathcal{U}| \times 2}$ and $\boldsymbol{o} \in \mathbb{R}^{|\mathcal{L}|}$. The output is then given as $f((x_j, x_k); \boldsymbol{W}, \boldsymbol{o}) = \sum_{l \in \mathcal{L}} P(\{(x_j, x_k) \to l\}) o_l$ where $P(\{(x_j, x_k) \to l\})$ is the proportion of $(x_j, x_k)$ that is routed to leaf $l$. Particularly, $P(\{(x_j, x_k) \to l\}) = \prod_{i \in \mathcal{A}(l)} r_{i,l}(x_j, x_k)$ where $\mathcal{A}(l)$ denotes the set of ancestors of $l$ and $r_{i,l}(x_j, x_k)$ denotes the proportion of $(x_j, x_k)$ is routed to leaf $l$ from node $i$. These values are given as $r_{i,l}(x_j, x_k) = \phi(\boldsymbol{W}_i^T(x_j, x_k))$ if $l$ belongs to the left subtree of $i$, and $r_{i,l}(x_j, x_k) = 1 - \phi(\boldsymbol{W}_i^T(x_j, x_k))$ if $l$ belongs to the right subtree of $i$, where $\phi(\cdot)$ is the sigmoid activation function. A soft tree $f(\cdot; \boldsymbol{W}, \boldsymbol{o}) : \mathbb{R} \mapsto \mathbb{R}$ for a main effect is defined similarly.

## B    Smooth-Step Function

The smooth-step function [18] has been used in soft trees for conditional computation [18] and routing in sparse mixture of experts [19, 21]. It can output exact zeros and ones, thus allowing for hard selection. Let $\gamma$ be a non-negative scalar parameter. The smooth-step function takes the form:

$$h(t) = \begin{cases} 0 & \text{if } t \leq -\gamma/2 \\ -\frac{2}{\gamma^3} t^3 + \frac{3}{2\gamma} t + \frac{1}{2} & \text{if } \gamma/2 \leq t \leq \gamma/2 \\ 1 & \text{if } t \geq \gamma/2 \end{cases} \tag{B.1}$$

The smooth-step function is continuously differentiable, similar to the logistic function. Additionally, it performs hard selection, i.e., outside $[-\gamma/2, \gamma/2]$, the function produces exact zeros and ones.

## C    Proofs of Main Results

### C.1    Preliminaries

Before proceeding with the proof, we define some notation we use throughout this section. Given the data points $\boldsymbol{x}_1, \cdots, \boldsymbol{x}_n$, we let $P_n$ the empirical measure supported on $\boldsymbol{x}_1, \cdots, \boldsymbol{x}_n$, $P_n = \sum_{i=1}^n \delta_{\boldsymbol{x}_i}/n$. We define the norm with respect to $P_n$ as

$$\|f\|_{P_n} = \sqrt{\frac{1}{n} \sum_{i=1}^n f(\boldsymbol{x}_i)^2}. \tag{C.1}$$

Next, we define covering and packing numbers.

**Definition C.1** (Covering number [38]). Let $\mathcal{F}$ be a functional class and $\boldsymbol{x}_1, \cdots, \boldsymbol{x}_n$ be a set of $n$ observations. An $\epsilon$-cover of $\mathcal{F}$ with respect to $P_n$ is a set of functions $f_1, \cdots, f_N$ such that for any $f \in \mathcal{F}$, there exists $j \in [N]$ such that

$$\|f - f_j\|_{P_n} \leq \epsilon.$$

The covering number of $\mathcal{F}$ with respect to $P_n$ at scale $\epsilon$, denoted as $\mathcal{N}(\mathcal{F}, P_n, \epsilon)$, is the smallest $N$ such that there exists an $\epsilon$-cover of $\mathcal{F}$ with respect to $P_n$ with size $N$.

**Definition C.2** (Packing number [56]). Let $\mathcal{F}$ be a functional class and $\boldsymbol{x}_1, \cdots, \boldsymbol{x}_n$ be a set of $n$ observations. An $\epsilon$-packing of $\mathcal{F}$ with respect to $P_n$ is a set of functions $f_1, \cdots, f_N$ such that for any $i, j \in [N]$,

$$\|f_i - f_j\|_{P_n} \geq \epsilon.$$

The packing number of $\mathcal{F}$ with respect to $P_n$ at scale $\epsilon$, denoted as $\mathcal{P}(\mathcal{F}, P_n, \epsilon)$, is the largest $N$ such that there exists an $\epsilon$-packing of $\mathcal{F}$ with respect to $P_n$ with size $N$.

The following well-known lemma establishes the equivalence between packing and covering numbers.

**Lemma C.1** (Lemma 4.2.8 [56])**.** *For $\epsilon > 0$,*

$$\mathcal{P}(\mathcal{F}, P_n, 2\epsilon) \leq \mathcal{N}(\mathcal{F}, P_n, \epsilon) \leq \mathcal{P}(\mathcal{F}, P_n, \epsilon).$$

Finally, we define the fat-shattering dimension of a function class.

**Definition C.3** (Fat-shattering dimension [26])**.** Let $\mathcal{F}$ be a functional class. Given a subset $S = \{\boldsymbol{x}_1, \cdots, \boldsymbol{x}_n\} \subseteq \mathbb{R}^p$, the class $\mathcal{F}$ $\epsilon$-shatters $S$, if there exists $c = (c_1, \cdots, c_n) \in \mathbb{R}^n$, such that for every $\boldsymbol{e} \in \{0, 1\}^n$, there exists $f \in \mathcal{F}$ such that

$$f(\boldsymbol{x}_i) \geq c_i + \epsilon/2 \text{ for } e_i = 1, \text{ and } f(\boldsymbol{x}_i) \leq c_i - \epsilon/2 \text{ for } e_i = 0.$$

The fat-shattering dimension of $\mathcal{F}$ at scale $\epsilon$, denoted as $\mathsf{fat}_\epsilon(\mathcal{F})$ is the size of the largest subset $S$ that is $\epsilon$-shattered by $\mathcal{F}$.

The following theorem relates the fat-shattering dimension to the covering number discussed above.

**Theorem C.1.** *If there exists $v > 0$ such that*

$$\mathsf{fat}_{x\epsilon}(\mathcal{F}) \leq \frac{v}{x} \quad \forall x \geq 1,$$

*then*

$$\log \mathcal{N}(\mathcal{F}, P_n, C\epsilon) \leq Cv.$$

*Proof.* The theorem is a direct result of Theorems 6.3 of [50] and Lemma C.1. □

## C.2 Technical Lemmas

Let $\mathcal{U}, \mathcal{L}$ denote the set of internal nodes and leaves of a depth $d$ tree, respectively. Let us define the class of soft trees with depth $d$ and data dimension $p$ as

$$\mathcal{F}(d, p) = \left\{ f(\boldsymbol{x}; \boldsymbol{W}, \boldsymbol{o}) : B_p^2 \mapsto \mathbb{R}, \bar{u}(f(\boldsymbol{x}; \boldsymbol{W}, \boldsymbol{o})) \leq B \right\} \tag{C.2}$$

where $B_p^2$ is the Euclidean ball of dimension $p$, $\boldsymbol{W} \in \mathbb{R}^{|\mathcal{U}| \times p}$ and $\boldsymbol{o} \in \mathbb{R}^{|\mathcal{L}|}$ and

$$\bar{u}(f(\boldsymbol{x}; \boldsymbol{W}, \boldsymbol{o})) = \max_{k \in \mathcal{U}} \|\boldsymbol{w}_k\|_2 \vee \max_{l \in \mathcal{L}} |o_l|$$

with $\boldsymbol{w}_k \in \mathbb{R}^p$ being the vector of weights for node $k$ from $\boldsymbol{W}$. Our first result is characterizing the fat-shattering dimension of $\mathcal{F}(d, p)$.

**Lemma C.2.** *Let $A = 2 \vee 2^{d+2}B \vee B^2 dL2^{d+3}$. For $\epsilon > 0$,*

$$\mathsf{fat}_\epsilon(\mathcal{F}(d, p)) \leq \left( \frac{A}{\epsilon} \right)^p. \tag{C.3}$$

*Proof.* First, suppose $\epsilon > 2B2^d$. We show that $\mathsf{fat}_\epsilon(\mathcal{F}(d, p)) = 0$. To this end, note that for $\boldsymbol{x}$ and $f \in \mathcal{F}(d, p)$,

$$|f(\boldsymbol{x})| = \left| \sum_{l \in \mathcal{L}} P(\{\boldsymbol{x} \to l\}) o_l \right| \leq \sum_{l \in \mathcal{L}} |o_l| \leq 2^d B \tag{C.4}$$

where $P$ is defined in Appendix A. As a result, for $f_1, f_2 \in \mathcal{F}(d, p)$ and $\boldsymbol{x}$,

$$|f_1(\boldsymbol{x}) - f_2(\boldsymbol{x})| \leq 2B2^d < \epsilon \tag{C.5}$$

showing no single point can be $\epsilon$-shattered. Therefore, $\mathsf{fat}_\epsilon(\mathcal{F}(d, p)) = 0$.
Let $0 < \epsilon \leq 2B2^d$ and suppose $\{\boldsymbol{x}_1, \cdots, \boldsymbol{x}_N\}$ is $\epsilon$-shattered by $\mathcal{F}(d, p)$ at level $\boldsymbol{c}$. We show that for $i, j \in [N]$, $\|\boldsymbol{x}_i - \boldsymbol{x}_j\|_2 \geq \epsilon/(B^2 dL2^d)$. To this end, assume there exists $i, j \in [N]$ such that $\|\boldsymbol{x}_i - \boldsymbol{x}_j\|_2 < \epsilon/(B^2 dL2^d)$. Without loss of generality assume $c_i \geq c_j$ and choose $\boldsymbol{e} \in \{0, 1\}^p$ such that $e_i = 1, e_j = 0$. Suppose $f(\cdot; \boldsymbol{W}, \boldsymbol{o}) \in \mathcal{F}(d, p)$ shatters the data with $\boldsymbol{e}$. Particularly,

$$f(\boldsymbol{x}_i; \boldsymbol{W}, \boldsymbol{o}) \geq c_i + \epsilon/2, \quad f(\boldsymbol{x}_j; \boldsymbol{W}, \boldsymbol{o}) \leq c_j - \epsilon/2. \tag{C.6}$$

For $k \in \mathcal{U}, l \in \mathcal{L}$, if $k \in \mathcal{A}(l)$ we have

$$|r_{k,l}(\boldsymbol{x}_i) - r_{k,l}(\boldsymbol{x}_j)| = |\phi(\boldsymbol{w}_k^T \boldsymbol{x}_i) - \phi(\boldsymbol{w}_k^T \boldsymbol{x}_i)|$$
$$\overset{(a)}{\leq} L|\boldsymbol{w}_k^T \boldsymbol{x}_i - \boldsymbol{w}_k^T \boldsymbol{x}_j|$$
$$\overset{(b)}{\leq} LB\|\boldsymbol{x}_i - \boldsymbol{x}_j\|_2 \tag{C.7}$$

where $r_{k,l}$ is defined in Appendix A, $\boldsymbol{w}_k \in \mathbb{R}^p$ is the vector of weights for node $k$ from $\boldsymbol{W}$, $(a)$ is by (A1) and $(b)$ is by $\bar{u}(f(\boldsymbol{x}; \boldsymbol{W}, \boldsymbol{o})) \leq B$ (recall the Assumption (A2)). Next,

$$|P(\{\boldsymbol{x}_i \to l\}) - P(\{\boldsymbol{x}_j \to l\})| = \left| \prod_{k \in \mathcal{A}(l)} r_{k,l}(\boldsymbol{x}_i) - \prod_{k \in \mathcal{A}(l)} r_{k,l}(\boldsymbol{x}_j) \right|$$
$$\leq dLB\|\boldsymbol{x}_i - \boldsymbol{x}_j\|_2 \tag{C.8}$$

where the inequality is by (C.7) and the fact that $r_{k,l} \in [0,1]$ is bounded. Finally,

$$|f(\boldsymbol{x}_i; \boldsymbol{W}, \boldsymbol{o}) - f(\boldsymbol{x}_j; \boldsymbol{W}, \boldsymbol{o})| = \left| \sum_{l \in \mathcal{L}} P(\{\boldsymbol{x}_i \to l\})o_l - \sum_{l \in \mathcal{L}} P(\{\boldsymbol{x}_j \to l\})o_l \right|$$
$$\leq \sum_{l \in \mathcal{L}} |o_l| |P(\{\boldsymbol{x}_i \to l\}) - P(\{\boldsymbol{x}_j \to l\})|$$
$$\leq 2^d B^2 dL \|\boldsymbol{x}_i - \boldsymbol{x}_j\|_2 \tag{C.9}$$

where the last inequality is true as $|\mathcal{L}| = 2^d$ and $|o_l| \leq B$ by $\bar{u}(f(\boldsymbol{x}; \boldsymbol{W}, \boldsymbol{o})) \leq B$. As we assumed $\|\boldsymbol{x}_i - \boldsymbol{x}_j\|_2 < \epsilon/(B^2 dL2^d)$, we have from (C.9)

$$|f(\boldsymbol{x}_i; \boldsymbol{W}, \boldsymbol{o}) - f(\boldsymbol{x}_j; \boldsymbol{W}, \boldsymbol{o})| < \epsilon$$

while on the other hand, by the shattering property (C.6),

$$f(\boldsymbol{x}_i; \boldsymbol{W}, \boldsymbol{o}) - f(\boldsymbol{x}_j; \boldsymbol{W}, \boldsymbol{o}) \geq \underbrace{(c_i - c_j)}_{\geq 0} + 2\epsilon/2 \geq \epsilon \tag{C.10}$$

which is a contradiction. Therefore, for every $i, j \in [N]$, we must have $\|\boldsymbol{x}_i - \boldsymbol{x}_j\|_2 \geq \epsilon/(B^2 dL2^d)$ or in other words, $\{\boldsymbol{x}_1, \cdots, \boldsymbol{x}_N\}$ is a $\epsilon/(B^2 dL2^d)$-packing of $B_p^2$. Thus,

$$N \overset{(a)}{\leq} \mathcal{N}(B_p^2, \|\cdot\|_2, \epsilon/(2B^2 dL2^d))$$
$$\overset{(b)}{\leq} \left( \frac{4B^2 dL2^d}{\epsilon} + 1 \right)^p$$
$$\leq \left( \frac{4B^2 dL2^d}{\epsilon} + 1 \vee 2B2^d \right)^p$$
$$\overset{(c)}{\leq} \left( \frac{4B^2 dL2^d}{\epsilon} + \frac{1 \vee 2B2^d}{\epsilon} \right)^p \leq \left( \frac{A}{\epsilon} \right)^p$$

where $\mathcal{N}(B_p^2, \|\cdot\|_2, \epsilon)$ is the $\epsilon$-covering number of $B_p^2$, $(a)$ is by Lemma C.1, $(b)$ is true by Corollary 4.2.13 of [56] and $(c)$ is true as $\epsilon \leq 2^{d+1}B$.

$\square$

**Lemma C.3.** *There exists an absolute constant $C > 0$ such that for $\epsilon > 0$,*

$$\log \mathcal{N}_p(\mathcal{F}(d,p), P_n, \epsilon) \leq C \left( \frac{CA}{\epsilon} \right)^p.$$

*Proof.* From Lemma C.2, for $x \geq 1$

$$\mathsf{fat}_{x\epsilon}(\mathcal{F}(d,p)) \leq \left(\frac{A}{x\epsilon}\right)^p$$

$$= \left(\frac{A}{\epsilon}\right)^p \frac{1}{x^p}$$

$$\leq \left(\frac{A}{\epsilon}\right)^p \frac{1}{x} \tag{C.11}$$

where the last inequality is true as $p \geq 1$. The proof is complete by Theorem C.1. $\qquad\square$

In the next step, we consider trees that only use a subset of variables, such as $\mathcal{J} \subseteq [p]$. Formally, we let

$$\mathcal{F}_0(\mathcal{J},d) = \left\{ f(\{x_j\}_{j\in\mathcal{J}}; \boldsymbol{W}, \boldsymbol{o}) : B_p^2 \mapsto \mathbb{R}, , \bar{u}(f(\{x_j\}_{j\in\mathcal{J}}; \boldsymbol{W}, \boldsymbol{o})) \leq B \right\}. \tag{C.12}$$

**Lemma C.4.** *For $\epsilon > 0$,*

$$\log \mathcal{N}(\mathcal{F}_0(\mathcal{J},d), P_n, \epsilon) \leq C \left(\frac{CA}{\epsilon}\right)^{|\mathcal{J}|}.$$

*Proof.* Let $P_{n,\mathcal{J}}$ be the empirical distribution supported on coordinates in $\mathcal{J}$,

$$P_{n,\mathcal{J}} = \frac{1}{n} \sum_{i=1}^{n} \delta_{\boldsymbol{x}_{\mathcal{J}}}$$

where $\boldsymbol{x}_{\mathcal{J}}$ is the sub-vector of $\boldsymbol{x}$ with coordinates in $\mathcal{J}$. Then,

$$\mathcal{N}(\mathcal{F}_0(\mathcal{J},d), P_n, \epsilon) \leq \mathcal{N}(\mathcal{F}(d,|\mathcal{J}|), P_{n,\mathcal{J}}, \epsilon)$$

The result follows from Lemma C.3. $\qquad\square$

Next, let us define the classes of functions with interactions in $\mathbb{J} \subseteq 2^{[p]}$,

$$\mathcal{F}_1(d,\mathbb{J}) = \left\{ \sum_{\mathcal{J}\in\mathbb{J}} f_{\mathcal{J}}(\boldsymbol{x}; \boldsymbol{W}_{\mathcal{J}}, \boldsymbol{o}_{\mathcal{J}}) : B_p^2 \mapsto \mathbb{R}, f_{\mathcal{J}}(\cdot; \boldsymbol{W}_{\mathcal{J}}, \boldsymbol{o}_{\mathcal{J}}) \in \mathcal{F}_0(\mathcal{J},d) \right\} \tag{C.13}$$

and the class of functions with at most $s$ interactions of size at most $t$ (i.e $t$-dimensional interactions),

$$\mathcal{F}^*(d,t,s) = \left\{ f_1 + f_2 : \mathbb{J} \subseteq 2^{[p]}, |\mathbb{J}| \leq s, |\mathcal{J}| \leq t \,\forall\mathcal{J} \in \mathbb{J}, \; f_1 \in \mathcal{F}_1(d,\mathbb{J}), \; f_2 \in \mathcal{F}_1(d,\mathbb{J}^*) \right\} \tag{C.14}$$

for a fixed $\mathbb{J}^*$ with $|\mathbb{J}^*| \leq s$ and $|\mathcal{J}| \leq t$ for $\mathcal{J} \in \mathbb{J}^*$.

**Lemma C.5.** *Suppose $0 < \epsilon \leq 2sCA$ and $1 \leq t \leq 2$. Then,*

$$\log \mathcal{N}(\mathcal{F}^*(d,t,s), P_n, \epsilon) \leq 2s \log p + 2Cs \left(\frac{2CsA}{\epsilon}\right)^t.$$

*Proof.* Let $\mathcal{S}(\mathcal{F}_0(\mathcal{J},d))$ be a minimal $\epsilon/2s$-cover of $\mathcal{F}_0(\mathcal{J},d)$. Hence, $|\mathcal{S}(\mathcal{F}_0(\mathcal{J},d))| = \mathcal{N}(\mathcal{F}_0(\mathcal{J},d), P_n, \epsilon/2s)$. Define

$$\mathbb{S} = \left\{ \mathbb{J} \subseteq 2^{[p]}, |\mathcal{J}| \leq t \,\forall\mathcal{J} \in \mathbb{J}, |\mathbb{J}| \leq s \right\} \tag{C.15}$$

and

$$\mathcal{S}^* = \bigcup_{\mathbb{J}\in\mathbb{S}} \left( \sum_{\mathcal{J}\in\mathbb{J}} \mathcal{S}(\mathcal{F}_0(\mathcal{J},d)) + \sum_{\mathcal{J}\in\mathbb{J}^*} \mathcal{S}(\mathcal{F}_0(\mathcal{J},d)) \right) \tag{C.16}$$

where $A + B = \{a + b : a \in A, b \in B\}$ denotes the Minkowski sum. Note that from Lemma C.4, for $\mathbb{J} \in \mathbb{S}$ and $\mathcal{J} \in \mathbb{J}$,

$$\mathcal{N}(\mathcal{F}_0(\mathcal{J}, d), P_n, \epsilon/2s) \leq \exp\left(C\left(\frac{2sCA}{\epsilon}\right)^{|\mathcal{J}|}\right) \leq \exp\left(C\left(\frac{2sCA}{\epsilon}\right)^t\right) \tag{C.17}$$

as $2sCA/\epsilon \geq 1$. For the rest of the proof, we let

$$N_t(\epsilon) = \exp\left(C\left(\frac{CA}{\epsilon}\right)^t\right).$$

Suppose $f \in \mathcal{F}^*(d, t, s)$,

$$f(\boldsymbol{x}) = \sum_{\mathcal{J} \in \mathbb{J}_0} f_{\mathcal{J}}(\boldsymbol{x}; \boldsymbol{W}_{\mathcal{J}}, \boldsymbol{o}_{\mathcal{J}}) + \sum_{\mathcal{J} \in \mathbb{J}^*} f_{\mathcal{J}}(\boldsymbol{x}; \boldsymbol{W}_{\mathcal{J}}, \boldsymbol{o}_{\mathcal{J}}) \tag{C.18}$$

for some $\mathbb{J}_0 \in \mathbb{S}$. For any $\mathcal{J} \in \mathbb{J}_0 \cup \mathbb{J}^*$, let $\tilde{f}_{\mathcal{J}} \in \mathcal{S}(\mathcal{F}_0(\mathcal{J}, d))$ be such that $\|\tilde{f}_{\mathcal{J}} - f_{\mathcal{J}}\|_{P_n} \leq \epsilon/2s$. Let

$$\tilde{f}(\boldsymbol{x}) = \sum_{\mathcal{J} \in \mathbb{J}_0} \tilde{f}_{\mathcal{J}}(\boldsymbol{x}; \tilde{\boldsymbol{W}}_{\mathcal{J}}, \tilde{\boldsymbol{o}}_{\mathcal{J}}) + \sum_{\mathcal{J} \in \mathbb{J}^*} \tilde{f}_{\mathcal{J}}(\boldsymbol{x}; \tilde{\boldsymbol{W}}_{\mathcal{J}}, \tilde{\boldsymbol{o}}_{\mathcal{J}}). \tag{C.19}$$

Note that $\tilde{f} \in \mathcal{S}^*$. Next,

$$\begin{aligned}
\|f - \tilde{f}\|_{P_n} &= \left\|\sum_{\mathcal{J} \in \mathbb{J}_0}(f_{\mathcal{J}} - \tilde{f}_{\mathcal{J}}) + \sum_{\mathcal{J} \in \mathbb{J}^*}(f_{\mathcal{J}} - \tilde{f}_{\mathcal{J}})\right\|_{P_n} \\
&\leq \sum_{\mathcal{J} \in \mathbb{J}_0}\|f_{\mathcal{J}} - \tilde{f}_{\mathcal{J}}\|_{P_n} + \sum_{\mathcal{J} \in \mathbb{J}^*}\|f_{\mathcal{J}} - \tilde{f}_{\mathcal{J}}\|_{P_n} \\
&\leq 2s\epsilon/2s = \epsilon \tag{C.20}
\end{aligned}$$

as $|\mathbb{J}_0|, |\mathbb{J}^*| \leq s$. As a result, $\mathcal{S}^*$ is a $\epsilon$-cover for $\mathcal{F}^*(d, t, s)$. Moreover,

$$\begin{aligned}
\mathcal{N}(\mathcal{F}^*(d, t, s), P_n, \epsilon) &\leq |\mathcal{S}^*| \\
&\leq \sum_{\mathbb{J} \in \mathbb{S}} \prod_{\mathcal{J} \in \mathbb{J}} |\mathcal{S}(\mathcal{F}_0(\mathcal{J}, d))| \prod_{\mathcal{J} \in \mathbb{J}^*} |\mathcal{S}(\mathcal{F}_0(\mathcal{J}, d))| \\
&\leq \sum_{\mathbb{J} \in \mathbb{S}} \prod_{\mathcal{J} \in \mathbb{J}} \mathcal{N}(\mathcal{F}_0(\mathcal{J}, d), P_n, \epsilon/2s) \prod_{\mathcal{J} \in \mathbb{J}^*} \mathcal{N}(\mathcal{F}_0(\mathcal{J}, d), P_n, \epsilon/2s) \\
&\overset{(a)}{\leq} \sum_{\mathbb{J} \in \mathbb{S}} N_t(\epsilon/2s)^{2s} \\
&\overset{(b)}{\leq} p^{2s} N_t(\epsilon/2s)^{2s} \tag{C.21}
\end{aligned}$$

where $(a)$ is by (C.17) and $(b)$ is true as $|\mathbb{S}| \leq (p^t)^s \leq p^{2s}$. As a result,

$$\begin{aligned}
\log \mathcal{N}(\mathcal{F}^*(d, t, s), P_n, \epsilon) &\leq 2s \log p + 2s \log N_t(\epsilon/2s) \\
&\leq 2s \log p + 2Cs\left(\frac{2CsA}{\epsilon}\right)^t. \tag{C.22}
\end{aligned}$$

$$\square$$

**Lemma C.6.** *Suppose* $0 < \delta \leq 1 \wedge 2sCA$. *Then, for* $a = \exp(-1)$,

$$\begin{aligned}
\int_{\delta^2/4\sigma}^{\delta} \sqrt{\log \mathcal{N}(\mathcal{F}^*(d, 1, s), P_n, \epsilon)} d\epsilon &\leq (\sqrt{2s \log p} + CsA)\sqrt{\delta} \\
\int_{\delta^2/4\sigma}^{\delta} \sqrt{\log \mathcal{N}(\mathcal{F}^*(d, 2, s), P_n, \epsilon)} d\epsilon &\leq \left(\sqrt{2s \log p} + \sqrt{2Cs}(2CsA)\right)(4\sigma/\delta)^a. \tag{C.23}
\end{aligned}$$

*Proof.* Note that if $\delta > 4\sigma$, the lemma is trivial. Therefore, we assume $\delta \leq 4\sigma$. From Lemma C.5

$$\sqrt{\log \mathcal{N}(\mathcal{F}^*(d,t,s), P_n, \epsilon)} \leq \sqrt{2s \log p + 2Cs \left( \frac{2CsA}{\epsilon} \right)^t}$$

$$\leq \sqrt{2s \log p} + \sqrt{2Cs}(2CsA)^{t/2} \epsilon^{-t/2}. \qquad (C.24)$$

Hence, as $\delta \leq 1$,

$$\int_{\delta^2/4\sigma}^{\delta} \sqrt{\log \mathcal{N}(\mathcal{F}^*(d,1,s), P_n, \epsilon)} d\epsilon \leq \int_0^{\delta} \sqrt{\log \mathcal{N}(\mathcal{F}^*(d,1,s), P_n, \epsilon)} d\epsilon$$

$$\leq \int_0^{\delta} \sqrt{2s \log p} d\epsilon + \int_0^{\delta} 2CsA\epsilon^{-1/2} d\epsilon$$

$$\leq \sqrt{2s \log p}\delta + CsA\delta^{1/2}$$

$$\leq \sqrt{2s \log p}\delta^{1/2} + CsA\delta^{1/2} \qquad (C.25)$$

Moreover,

$$\int_{\delta^2/4\sigma}^{\delta} \sqrt{\log \mathcal{N}(\mathcal{F}^*(d,2,s), P_n, \epsilon)} d\epsilon \leq \int_{\delta^2/4\sigma}^{\delta} \sqrt{\log \mathcal{N}(\mathcal{F}^*(d,2,s), P_n, \epsilon)} d\epsilon$$

$$\leq \int_0^{\delta} \sqrt{2s \log p} d\epsilon + \int_{\delta^2/4\sigma}^{\delta} 2CsA\sqrt{2Cs}\epsilon^{-1} d\epsilon$$

$$\leq \sqrt{2s \log p} + 2\sqrt{2Cs}CsA \log(4\sigma/\delta). \qquad (C.26)$$

Next, consider the function $h(x) = x^a - \log(x)$ where $a = \exp(-1)$. We will show $h(x) \geq 0$ for $x > 0$. Note that

$$h'(x) = ax^{a-1} - \frac{1}{x}$$

therefore $h(a^{-1/a}) = h'(a^{-1/a}) = 0$. Moreover,

$$h''(x) = \frac{a(a-1)x^a + 1}{x^2}$$

so $h''(a^{-1/a}) = (1 + a(a-1)a^{-1})/(a^{-2/a}) > 0$, showing that $a^{-1/a}$ is a minimum for $h(x)$, and therefore $h(x) \geq 0$ for $x > 0$. Hence, from (C.26),

$$\int_{\delta^2/4\sigma}^{\delta} \sqrt{\log \mathcal{N}(\mathcal{F}^*(d,2,s), P_n, \epsilon)} d\epsilon \leq \sqrt{2s \log p} + \sqrt{2Cs}2CsA \log(4\sigma/\delta)$$

$$\leq \sqrt{2s \log p} + \sqrt{2Cs}2CsA(4\sigma/\delta)^a$$

$$\leq \sqrt{2s \log p}(4\sigma/\delta)^a + \sqrt{2Cs}2CsA(4\sigma/\delta)^a \qquad (C.27)$$

as $\delta \leq 4\sigma$ or $4\sigma/\delta \geq 1$. $\qquad \square$

**Lemma C.7.** *Under the assumptions of Theorem 2, for any sequence $a_n, n \geq 1$ such that $a_n > 0$ and $\lim_{n \to \infty} a_n = 0$, there exists a positive sequence $b_n$ such that $\lim_{n \to \infty} b_n = 0$ and*

$$\int_{\delta^2/4\sigma}^{\delta} \sqrt{\log \mathcal{N}(\mathcal{F}^*(d,2,s), P_n, \epsilon)} d\epsilon \leq \left( \sqrt{2s \log p} + \sqrt{2Cs}(2CsA) \right) (4\sigma/\delta)^{a_n} \quad \forall \delta \leq b_n.$$

*Proof.* Consider the function $h_n(x) = x^{a_n} - \log(x)$. Note that

$$\lim_{x \to \infty} \frac{x^{a_n}}{\log x} = \lim_{x \to \infty} a_n x^{a_n} = \infty$$

therefore, there exists a sequence $\gamma_n$ such that $\lim_{n \to \infty} \gamma_n = \infty$ and

$$h_n(x) \geq 0, \quad \forall x \geq \gamma_n.$$

Therefore, if $\delta \leq 4\sigma/\gamma_n \wedge 4\sigma \wedge 1 =: b_n$, from (C.26),

$$\int_{\delta^2/4\sigma}^{\delta} \sqrt{\log \mathcal{N}(\mathcal{F}^*(d,2,s), P_n, \epsilon)} d\epsilon \leq \sqrt{2s\log p} + 2\sqrt{2Cs}CsA\log(4\sigma/\delta)$$

$$\leq \sqrt{2s\log p} + 2\sqrt{2Cs}CsA(4\sigma/\delta)^{a_n}$$

$$\leq \left(\sqrt{2s\log p} + \sqrt{2Cs}(2CsA)\right)(4\sigma/\delta)^{a_n}. \quad \text{(C.28)}$$

$\square$

**Lemma C.8.** *For any $\epsilon, \delta > 0$,*

$$\mathcal{N}(\mathcal{F}^*(d,t,s) \cap \{\|f\|_{P_n} \leq \delta\}, P_n, \epsilon) \leq \mathcal{N}(\mathcal{F}^*(d,t,s), P_n, \epsilon). \quad \text{(C.29)}$$

*Proof.* Let $\mathcal{S} = \{f_1, \cdots, f_N\}$ be a minimal $\epsilon$-cover of $\mathcal{F}^*(d,t,s)$ where $N = \mathcal{N}(\mathcal{F}^*(d,t,s), P_n, \epsilon)$. Let $\mathcal{S}^* = \{f_1^*, \cdots, f_N^*\}$ where for $i \in [N]$,

$$f_i^* = \begin{cases} f_i & \text{if } \|f_i\|_{P_n} \leq \delta \\ f_i\delta/\|f_i\|_{P_n} & \text{if } \|f_i\|_{P_n} > \delta \end{cases} \in \mathcal{F}^*(d,t,s) \cap \{\|f\|_{P_n} \leq \delta\}.$$

We show $\mathcal{S}^*$ is a $\epsilon$-cover for $\mathcal{F}^*(d,t,s) \cap \{\|f\|_{P_n} \leq \delta\}$.
Let $f \in \mathcal{F}^*(d,t,s) \cap \{\|f\|_{P_n} \leq \delta\}$. Let $j \in [N]$ be such that $\|f - f_j\|_{P_n} \leq \epsilon$. Let us consider the following cases:
**Case 1:** If $\|f_j\|_{P_n} \leq \delta$, then $f_j \in \mathcal{S}^*$ showing $f_j^* = f_j$ covers $f$.
**Case 2:** Suppose $\|f_j\|_{P_n} > \delta$. For any $f$, let $\mathbf{f} = (f(\mathbf{x}_1), \cdots, f(\mathbf{x}_n))/\sqrt{n} \in \mathbb{R}^n$. With this notation, $\|f\|_{P_n} = \|\mathbf{f}\|_2$. As a result,

$$\begin{aligned}
\epsilon^2 &\geq \|f - f_j\|_{P_n}^2 \\
&= \left\|(f - f_j^*) + (f_j^* - f_j)\right\|_{P_n}^2 \\
&= \left\|(f - f_j^*) + \left(\frac{\delta}{\|f_j\|_{P_n}} - 1\right)f_j\right\|_{P_n}^2 \\
&= \left\|(\mathbf{f} - \mathbf{f}_j^*) + \left(\frac{\delta}{\|\mathbf{f}_j\|_2} - 1\right)\mathbf{f}_j\right\|_2^2 \\
&\geq \|\mathbf{f} - \mathbf{f}_j^*\|_2^2 + 2\left(\frac{\delta}{\|\mathbf{f}_j\|_2} - 1\right)\mathbf{f}_j^T(\mathbf{f} - \mathbf{f}_j^*) \\
&= \|\mathbf{f} - \mathbf{f}_j^*\|_2^2 + 2\left(\frac{\delta}{\|\mathbf{f}_j\|_2} - 1\right)\mathbf{f}_j^T\mathbf{f} - 2\left(\frac{\delta}{\|\mathbf{f}_j\|_2} - 1\right)\frac{\delta}{\|\mathbf{f}_j\|_2}\|\mathbf{f}_j\|_2^2 \\
&\overset{(a)}{\geq} \|\mathbf{f} - \mathbf{f}_j^*\|_2^2 + 2\left(\frac{\delta}{\|\mathbf{f}_j\|_2} - 1\right)\|\mathbf{f}_j\|_2\delta - 2\left(\frac{\delta}{\|\mathbf{f}_j\|_2} - 1\right)\delta\|\mathbf{f}_j\|_2 \\
&= \|\mathbf{f} - \mathbf{f}_j^*\|_2^2 = \|f - f_j^*\|_{P_n}^2 \quad \text{(C.30)}
\end{aligned}$$

where $(a)$ is true as $\|\mathbf{f}_j\|_2 > \delta$ and $\|\mathbf{f}\|_2 \leq \delta$. This shows $f_j^*$ covers $f$, completing the proof. $\square$

## C.3 Proof of Theorem 1

*Proof.* Let

$$\delta_1 = \frac{\left[64\sigma(\sqrt{2s\log p} + CsA)\right]^{2/3}}{n^{1/3}}$$

$$\delta_2 = \frac{\left[16\left(\sqrt{2s\log p} + \sqrt{2Cs}(2CsA)\right)(4\sigma)^{1+a}\right]^{1/(2+a)}}{n^{1/2(2+a)}}. \quad \text{(C.31)}$$

**Claim:** For $t \in \{1, 2\}$,

$$\frac{16}{\sqrt{n}}\int_{\delta_t^2/4\sigma}^{\delta_t} \sqrt{\log \mathcal{N}(\mathcal{F}^*(d,t,s) \cap \{\|f\|_{P_n} \leq \delta_t\}, P_n, \epsilon)} d\epsilon \leq \frac{\delta_t^2}{4\sigma}.$$

We will prove the claim later. Let

$$\mathcal{F}(d,t,s) = \left\{ f_1 : \mathbb{J} \subseteq 2^{[p]}, |\mathbb{J}| \leq s, |\mathcal{J}| \leq t \; \forall \mathcal{J} \in \mathbb{J}, \; f_1 \in \mathcal{F}_1(d,\mathbb{J}) \right\}. \tag{C.32}$$

Under this notation, we have

$$\hat{f}_t \in \operatorname*{argmin}_{f \in \mathcal{F}(d,t,s)} \frac{1}{n} \sum_{i=1}^{n} (y_i - f(\boldsymbol{x}_i))^2.$$

Using the notation from [57][Ch. 13], we have

$$\partial \mathcal{F}(d,t,s) = \mathcal{F}(d,t,s) - \mathcal{F}(d,t,s) = \mathcal{F}^*(d,t,s)$$

with $\mathbb{J}^*$ being the set of interactions for for $f_t^*$. Moreover, we note that $\mathcal{F}^*(d,t,s)$ is star-shaped, that is for $f \in \mathcal{F}^*(d,t,s)$ and $\alpha \in [0,1]$, $\alpha f \in \mathcal{F}^*(d,t,s)$. This is true as

$$\alpha f(\cdot; \boldsymbol{W}, \boldsymbol{o}) = f(\cdot; \boldsymbol{W}, \alpha \boldsymbol{o}).$$

Under the claim and assumptions of the theorem, we have $\delta_t \leq \sigma$ so we invoke Corollary 13.7 and Theorem 13.13 of [57] with $\gamma = 1/2, t = \delta$ to achieve

$$\mathbb{P}\left( \frac{1}{n} \sum_{i=1}^{n} (y_i^* - \hat{f}_t(\boldsymbol{x}_i))^2 \lesssim \inf_{f \in \mathcal{F}(d,t,s)} \sum_{i=1}^{n} (y_i^* - f(\boldsymbol{x}_i))^2 + \delta_t^2 \right) \geq 1 - c_1 \exp(-c_2 n \delta_t^2 / \sigma^2) \tag{C.33}$$

completing the proof.

**Proof of Claim:** For $t = 1$,

$$\begin{aligned}
\frac{\delta_1^2}{4\sigma} &= \frac{\left[ 64\sigma(\sqrt{2s\log p} + CsA) \right]^{4/3}}{4\sigma n^{2/3}} \\
&= \frac{16}{\sqrt{n}} (\sqrt{2s\log p} + CsA) \frac{\left[ 64\sigma(\sqrt{2s\log p} + CsA) \right]^{1/3}}{n^{1/6}} \\
&= \frac{16}{\sqrt{n}} (\sqrt{2s\log p} + CsA) \sqrt{\delta_1} \\
&\geq \frac{16}{\sqrt{n}} \int_{\delta_1^2/4\sigma}^{\delta_1} \sqrt{\log \mathcal{N}(\mathcal{F}^*(d,1,s), P_n, \epsilon)} d\epsilon \\
&\geq \frac{16}{\sqrt{n}} \int_{\delta_1^2/4\sigma}^{\delta_1} \sqrt{\log \mathcal{N}(\mathcal{F}^*(d,1,s) \cap \{\|f\|_{P_n} \leq \delta_1\}, P_n, \epsilon)} d\epsilon \tag{C.34}
\end{aligned}$$

where the first inequality is by Lemma C.6 and the final inequality is by Lemma C.8. Next, for $t = 2$,

$$\begin{aligned}
\frac{\delta_2^2}{4\sigma} &= \frac{\left[ 16\left(\sqrt{2s\log p} + \sqrt{2Cs}(2CsA)\right)(4\sigma)^{1+a} \right]^{2/(2+a)}}{4\sigma n^{1/(2+a)}} \\
&= \frac{16}{\sqrt{n}} \left(\sqrt{2s\log p} + \sqrt{2Cs}(2CsA)\right) \frac{\left[ 16\left(\sqrt{2s\log p} + \sqrt{2Cs}(2CsA)\right)(4\sigma)^{1+a} \right]^{-a/(2+a)} (4\sigma)^a}{n^{-a/2(2+a)}} \\
&= \frac{16}{\sqrt{n}} \left(\sqrt{2s\log p} + \sqrt{2Cs}(2CsA)\right) (4\sigma/\delta)^a \\
&\geq \frac{16}{\sqrt{n}} \int_{\delta_2^2/4\sigma}^{\delta_2} \sqrt{\log \mathcal{N}(\mathcal{F}^*(d,2,s) \cap \{\|f\|_{P_n} \leq \delta_2\}, P_n, \epsilon)} d\epsilon \tag{C.35}
\end{aligned}$$

by Lemmas C.6 and C.8. $\qquad\square$

### C.4   Proof of Theorem 2

*Proof.* Let

$$\delta_2 = \frac{\left[ 16\left(\sqrt{2s\log p} + \sqrt{2Cs}(2CsA)\right)(4\sigma)^{1+a_n} \right]^{1/(2+a_n)}}{n^{1/2(2+a_n)}}.$$

By taking $n \gtrsim b_n^{-2(2+a_n)}$ and invoking Lemmas C.7 and C.8,

$$\frac{\delta_2^2}{4\sigma} \geq \frac{16}{\sqrt{n}} \int_{\delta_2^2/4\sigma}^{\delta_2} \sqrt{\log \mathcal{N}(\mathcal{F}^*(d, 2, s) \cap \{\|f\|_{P_n} \leq \delta_2\}, P_n, \epsilon)} d\epsilon.$$

The rest of the proof is similar to the proof of Theorem 1. $\qquad\square$

### C.5 Theory Discussion

First, we consider the case with only main effects, i.e $t = 1$. In our framework, we use the smooth-step function which is differentiable with uniformly bounded derivative, but not twice differentiable. Therefore, our framework operates on $L_2$ Sobolev functions of order 1, i.e. bounded differentiable functions with bounded gradients. For this case, the results of [52] show that an $\ell_2$ prediction rate of $n^{-2/3}$ (see Remark 4 of [52] with $q = 0$ and $\beta_0 = 1$). This is the same rate we achieve in Theorem 1 for $t = 1$. However, we note that our results hold without differentiability assumption and only require the activation function to be Lipschitz, showing stronger results specialized to the soft trees case.

Next, we consider the interaction case with $t = 2$. In this case, the results of [23] imply an $\ell_2$ prediction rate of $n^{-1/2}$ for the class of $L_2$ Sobolev functions of order 1 (see Section S2.2 with $m = 1$, leading to $r_n = n^{-1/4}$). This is the same rate our method achieves asymptotically (see Theorem 2). However, as we discussed, our results hold without the differentiability assumption.

## D  Extension to third-order interactions

Our approach can be extended to model third-order interactions. In the case of third-order interactions, the structural constraints for strong and weak hierarchy can be described as follows:

$$\text{Strong Hierarchy}: \quad f_{j,k,l} \neq 0 \implies f_j \neq 0 \text{ and } f_k \neq 0 \text{ and } f_l \neq 0, \qquad (D.1)$$
$$\text{Weak Hierarchy}: \quad f_{j,k,l} \neq 0 \implies f_j \neq 0 \text{ or } f_k \neq 0 \text{ or } f_l \neq 0. \qquad (D.2)$$

These constraints can be modeled with binary variables as follows:

$$\text{Strong Hierarchy}: \quad q(z_j, z_k, z_l, z_{j,k,l}) \stackrel{\text{def}}{=} z_j z_k z_l z_{j,k,l} \qquad (D.3)$$
$$\text{Weak Hierarchy}: \quad q(z_j, z_k, z_l, z_{j,k,l}) \stackrel{\text{def}}{=} (z_j + z_k + z_l - z_j z_k - z_j z_l - z_k z_l + z_j z_k z_l) z_{j,k,l} \qquad (D.4)$$

Hence, our sparse selection and hierarchy constraints can add value in terms of model compactness and feature selection in the settings with third-order interactions as well. However, for third-order interactions, the number of third-order interactions are $O(p^3)$. Hence, for scalability, this would also require using a pre-training screening approach. Despite the fact that our approach is generalizable to third-order interactions, we do not consider such interactions because these are not considered to be easily interpretable.

## E  Datasets, Computing Setup and Tuning

**Datasets**  We use a collection of 16 open-source classification datasets (binary, multiclass and regression) from various domains, e.g., physics, computing, healthcare, life sciences, finance, and social networks. They are from Penn Machine Learning Benchmarks (PMLB) [41] and UCI databases [8]. For datasets with available training, validation and test splits, we used them in their original form. When no test set was available, we treated the original validation set as the test set and split the training set into 80% training and 20% validation. For remaining, we randomly split each of the dataset into 60% training, 20% validation and 20% testing sets. A summary of the 16 datasets considered is in Table E.1.

**Computing Setup.** We used a cluster running Ubuntu 7.5.0 and equipped with Intel Xeon Platinum 8260 CPUs and Nvidia Volta V100 GPUs. For all experiments of Sec. 6, each job involving GRAND-SLAMIN, EBM, Node-GAM, SIAN, GAMI-Net and DNN were run on 8 core, 32GB RAM. Jobs involving larger datasets ($p > 100$) were run on Tesla V100 GPUs.

Table E.1: Summary of Datasets

| Dataset | Domain | $N$ | $C$ | $p$ | No. of interactions ($|\mathcal{I}|$) |
|---|---|---|---|---|---|
| Magic | Physics | 19,020 | 2 | 10 | 55 |
| Adult | Socio-economic | 48,842 | 2 | 14 | 91 |
| Churn | Business | 5,000 | 2 | 19 | 190 |
| Satimage | Physics | 6,435 | 6 | 36 | 640 |
| Texture | Image | 5,500 | 11 | 40 | 780 |
| MiniBooNE | Physics | 130,065 | 2 | 50 | 1,225 |
| Covertype | Life Science | 581,012 | 7 | 54 | 1,431 |
| Spambase | Computing | 4,601 | 2 | 57 | 1,596 |
| News | Social networks | 39,797 | 2 | 61 | 1,830 |
| Optdigits | Image | 5,620 | 10 | 64 | 2,016 |
| Bankruptcy | Finance | 6,819 | 2 | 96 | 4,560 |
| Madelon | NIPS-2003 | 2,600 | 2 | 500 | 124,750 |
| Activity | Healthcare | 4,480 | 4 | 533 | 141,778 |
| Multiple | Image | 2,000 | 10 | 649 | 210,276 |
| Bike Sharing | Transportation | 17,389 | - | 16 | 120 |
| American Community Survey | Demographic | 83,059 | - | 39 | 741 |

**Tuning.** The tuning was done in parallel over the competing models and datasets. We tune the hyperparameter using Optuna [2] which optimizes the overall AUC on a validation set. We report the results on a held-out test set. A list of all the tuning parameters and their distributions is given for `GRAND-SLAMIN` below:

- Learning Rates: Discrete uniform in the set $\{0.05, 0.01, 0.005\}$ for Adam with multi-step decay rate of $0.9$ every $25$ epochs.
- Batch-size: Discrete uniform in the set $\{64, 256\}$.
- $\lambda$ for selection: Discrete uniform in the set of 11 values $\{0, 1e-6, \cdots, 1e-3\}$.
- $\gamma$ for Smooth-step: Discrete uniform in the set $\{0.01, 0.1, 1\}$.
- $\tau$ for Entropy Regularization: Discrete uniform in the set $\{0.001, 0.01, 0.1\}$.
- $\alpha$ for relative penalty on interactions: Discrete uniform in the set $\{1, 10\}$.
- Epochs: 1000 with early stopping (patience=50) based on validation loss.
- For Madelon, Activity and Multiple datasets, we used screening to reduce the initial set of interactions to 5000.

For other toolkits, we considered the tuning protocols outlined in the respective papers such that the parameter controlling the variable selection is tuned. For SIAN, we tuned over the threshold parameter $\theta$ such that the screened set of interactions is upper bounded by $\{250, 500, \cdots, 1000\}$. We set $\tau = 1$ such that the model satisfies strong hierarchy. For GAMI-Net, we tuned over the number of interactions. For small datasets ($|\mathcal{I}| < 1000$), we tuned over the set: $\{0.2|\mathcal{I}|, 0.4|\mathcal{I}|, \cdot, |\mathcal{I}|\}$. For large datasets ($|\mathcal{I}| < 1000$), we tuned over the set: $\{250, 500, \cdots, 1000\}$. For EBM, we tuned over the set of interactions $\{16, 32, 64, 128\}$ as done by authors in [6]. For NODE-GA$^2$M, we tuned the number of trees as this controls the maximum number of interactions selected by the model.

## F  Additional Results

### F.1  Comparison with full complexity models e.g., DNN

Here, we compare interpretable modeling toolkits based on pairwise interactions with full complexity models. In particular, we consider a deep neural network (DNN). We considered the same architecture for the DNN as used by the authors in [10] for similar comparisons. The architecture is a 4-layered ReLU-activated neural network. We show the test AUC performance of different models in Table F.1. Across all datasets, we see that the full complexity DNN model underperforms interpretable models including `GRAND-SLAMIN`. However, it is noted that the choice of the architecture for DNN maybe contributing to the degradation in performance. There maybe other architecture choices for DNN e.g., neural networks with residual connections, that may perform better.

Table F.1: Test ROC AUC of GRAND-SLAMIN, EB$^2$M and NODE-GA$^2$M. We report median across 10 runs along with the mean absolute deviation (MAD).

| Dataset\Model | Interpretable Models | | | Full Complexity Models |
|---|---|---|---|---|
| | EB$^2$M | NODE-GA$^2$M | GRAND-SLAMIN | DNN |
| Magic | $93.12 \pm 0.001$ | $\mathbf{94.27} \pm 0.13$ | $93.86 \pm 0.30$ | $93.69 \pm 0.04$ |
| Adult | $91.41 \pm 0.0004$ | $\mathbf{91.75} \pm 0.14$ | $91.54 \pm 0.14$ | $90.26 \pm 0.04$ |
| Churn | $91.97 \pm 0.005$ | $89.62 \pm 5.61$ | $\mathbf{92.40} \pm 0.41$ | $90.28 \pm 0.34$ |
| Satimage | $97.65 \pm 0.001$ | $98.70 \pm 0.07$ | $\mathbf{98.81} \pm 0.04$ | $98.67 \pm 0.07$ |
| Texture | $99.81 \pm 0.0004$ | $\mathbf{100.00} \pm 0.00$ | $\mathbf{100.00} \pm 0.00$ | $99.63 \pm 0.09$ |
| MiniBooNE | $97.86 \pm 0.0001$ | $\mathbf{98.44} \pm 0.02$ | $97.77 \pm 0.05$ | $97.08 \pm 0.38$ |
| Covertype | $90.08 \pm 0.0003$ | $95.39 \pm 0.12$ | $\mathbf{98.11} \pm 0.08$ | $93.83 \pm 0.10$ |
| Spambase | $\mathbf{98.84} \pm 0.01$ | $98.78 \pm 0.06$ | $98.01 \pm 4.70$ | $98.09 \pm 0.07$ |
| News | $73.03 \pm 0.002$ | $\mathbf{73.53} \pm 0.06$ | $73.24 \pm 0.04$ | $72.19 \pm 0.08$ |
| Optdigits | $99.79 \pm 0.0003$ | $99.93 \pm 0.02$ | $\mathbf{99.98} \pm 0.00$ | $99.77 \pm 0.04$ |
| Bankruptcy | $\mathbf{93.85} \pm 0.01$ | $92.02 \pm 1.03$ | $92.51 \pm 0.54$ | $88.38 \pm 0.36$ |
| Madelon | $88.04 \pm 0.02$ | $60.07 \pm 0.82$ | $\mathbf{89.25} \pm 1.03$ | $63.60 \pm 0.47$ |
| Activity | $74.96 \pm 8.77$ | $\mathbf{99.86} \pm 0.04$ | $99.24 \pm 1.45$ | $96.64 \pm 1.93$ |
| Multiple | $\mathbf{99.96} \pm 0.0002$ | $99.94 \pm 0.02$ | $99.92 \pm 0.02$ | $99.94 \pm 0.01$ |

## F.2 GRAND-SLAMIN versus GAMs with all pairwise interactions

Next, we compare GRAND-SLAMIN i.e., GAMs with sparse pairwise interactions against GAMs with all pairwise interactions toolkit:

1. NA$^2$M, i.e., Neural Additive Model [1] with pairwise interactions,

2. NB$^2$M, i.e., Neural Bases Model [46] with pairwise interactions,

3. SPAM, i.e., Scalable Polynomial Additive Model [9] with pairwise interactions.

For NA$^2$M, NB$^2$M, and SPAM, we tuned over learning rate in the set $\{0.1, 0.01, 0.001, 0.0001\}$ and number of epochs in the set $\{50, 100, 500\}$. We capped the time for each trial for NA$^2$M to 6 hrs.

We show results in Table F.2. We outperform SPAM across many datasets. Notably, GRAND-SLAMIN improves by 21% over SPAM on Madelon. We are also competitive with NB$^2$M and NA$^2$M. For larger datasets e.g., Madelon, Activity and Multiple, NB$^2$M and NA$^2$M ran out of memory on a compute node with 2 V100 Tesla GPUs. Recall also that the goal of our work is to learn sparse components with/without structural constraints for easier interpretability.

Table F.2: Test ROC AUC for GRAND-SLAMIN and GAMs with all pairwise interaction models, e.g., NA$^2$M, NB$^2$M and SPAM.

| Dataset | NA$^2$M (Dense) | NB$^2$M (Dense) | SPAM (Dense) | GRAND-SLAMIN (Sparse) |
|---|---|---|---|---|
| Magic | $\mathbf{94.46} \pm 0.16$ | $94.11 \pm 0.08$ | $91.75 \pm 0.004$ | $93.86 \pm 0.30$ |
| Adult | $90.81 \pm 0.10$ | $91.06 \pm 0.03$ | $89.65 \pm 0.001$ | $\mathbf{91.54} \pm 0.14$ |
| Churn | $\mathbf{93.03} \pm 0.49$ | $92.16 \pm 0.42$ | $88.41 \pm 0.05$ | $92.40 \pm 0.41$ |
| Satimage | $98.81 \pm 0.05$ | $\mathbf{98.89} \pm 0.03$ | $97.94 \pm 0.02$ | $98.81 \pm 0.04$ |
| Covertype | out of time | $\mathbf{98.36} \pm 0.02$ | $96.29 \pm 0.02$ | $98.11 \pm 0.08$ |
| Spambase | $98.38 \pm 0.05$ | $98.37 \pm 0.06$ | $97.78 \pm 0.04$ | $\mathbf{98.55} \pm 0.07$ |
| News | $71.94 \pm 0.30$ | $72.54 \pm 0.07$ | $72.43 \pm 0.06$ | $\mathbf{73.24} \pm 0.04$ |
| Bankruptcy | $87.83 \pm 0.12$ | $\mathbf{93.01} \pm 1.85$ | $89.35 \pm 0.90$ | $92.51 \pm 0.54$ |
| Madelon | out of memory | out of memory | $68.59 \pm 0.80$ | $\mathbf{89.25} \pm 1.03$ |
| Activity | out of memory | out of memory | $99.10 \pm 0.04$ | $\mathbf{99.24} \pm 1.45$ |
| Multiple | out of memory | out of memory | $\mathbf{99.98} \pm 0.03$ | $99.92 \pm 0.02$ |

## F.3 Comparison with Group Lasso

Group Lasso [16] is also a possible choice for sparse selection, which is popularly used in high-dimensional statistics and machine learning. We compare against a version of Group Lasso:

$$\min_{\substack{\{f_i\}_{i \in [p]}, \\ \{f_{i,j}\}_{i<j}}} \hat{\mathbb{E}} \left[ l\left(y, \sum_{i \in [p]} f_i(x_i) + \sum_{i<j} f_{i,j}\right) \right] + \lambda \left( \sum_{i \in [p]} \|f_i\|_2 + \sum_{i<j} \|f_{i,j}\|_2 \right), \qquad \text{(F.1)}$$

where $\hat{\mathbb{E}}$ denotes the empirical loss on the training dataset and $\|f_i\|_2$ denotes the regularization imposed via the group regularization on the leaf weights of the effect $i$. We compare performance between GRAND-SLAMIN and Group Lasso (with soft tree shape functions) in terms of test AUC and variable selection on a few datasets. The numbers are reported in Table F.3. Overall, our approach significantly outperforms Group Lasso in terms of AUC performance and variable selection.

Table F.3: Comparison of Test ROC AUC of GRAND-SLAMIN with Group Lasso.

| Model Dataset | Group Lasso (AUC) | GRAND-SLAMIN (AUC) | Group Lasso (#features) | GRAND-SLAMIN (#features) |
|---|---|---|---|---|
| Adult | 91.19 | **91.54** $\pm$ 0.14 | 14 | **12** |
| Spambase | 98.32 | **98.81** $\pm$ 0.04 | 57 | **44** |
| Madelon | 65.22 | **90.13** $\pm$ 1.03 | 500 | **15** |

## F.4 GRAND-SLAMIN with different shape functions (Soft Trees vs MLPs)

Next, we compare GRAND-SLAMIN with different shape functions. In particular, we study the effect of using neural network shape functions instead of soft tree shape functions. We consider multilayer perceptrons (MLPs) as the functional form for each of the shape functions. We compare test AUC performance on 5 datasets in Table F.4. Interestingly, it seems to us that soft trees seem to have an edge over MLPs across some datasets, when MLPs are matched in number of parameters to soft trees.

Table F.4: Soft tree shape functions vs MLP shape functions.

| Dataset\Model | GRAND-SLAMIN with *MLPs* | GRAND-SLAMIN with *Soft Trees* |
|---|---|---|
| Magic | 93.13$\pm$0.12 | **93.86**$\pm$0.30 |
| Churn | 92.33$\pm$0.56 | **92.40**$\pm$0.41 |
| MiniBooNE | 97.41$\pm$0.21 | **97.77**$\pm$0.05 |
| Spambase | 98.27$\pm$0.13 | **98.55**$\pm$0.07 |
| News | 72.87$\pm$0.09 | **73.24**$\pm$0.04 |

**MLPs with varying complexity** We further experimented with MLPs in our framework, where we vary the complexity of the individual components. We show training time on News dataset ($p = 61$, $\#Interaction = 1800$, $N = 40k$) in Table F.5. The results show that the active set (learnt by learnable indicators) in the model is the primary contributing factor in terms of timing. The functions $(1 \text{ or } 2) \to 64 \to 64 \to 1$ have at least $22\times$ more parameters than $(1 \text{ or } 2) \to 64 \to 1$. However, the time only increases by $\sim 1.5\times$. We also show the AUC performance with MLPs with varying complexity in Table F.6. Interestingly, we didn't observe any improvement in AUCs with more complex components. It seems to us that since our additive modeling framework is fitting low-dimensional components — 1-dimensional functions for main effects and 2-dimensional functions for pairwise interactions effects — the complexity of the function class doesn't need to be too large to get good model accuracy.

## F.5 Effect of entropy regularization

Next, we study the impact of entropy regularization on performance and variable selection. It can be hypothesized that there might be a trade-off in the speed at which the binary state of the gate variables should be reached. On the one hand, suppression of the uninformative terms should happen fast as

Table F.5: Training times for different choices of MLP shape functions.

| % Components Selected | MLPs with varying complexity | Training times |
|---|---|---|
| 25% effects | $(1 \text{ or } 2) \rightarrow 64 \rightarrow 1$ | $252.4 \pm 0.4$ |
| | $(1 \text{ or } 2) \rightarrow 64 \rightarrow 64 \rightarrow 1$ | $247.6 \pm 0.8$ |
| | $(1 \text{ or } 2) \rightarrow 64 \rightarrow 64 \rightarrow 64 \rightarrow 1$ | $258.8 \pm 10.7$ |
| | $(1 \text{ or } 2) \rightarrow 128 \rightarrow 128 \rightarrow 1$ | $268.5 \pm 19.1$ |
| | $(1 \text{ or } 2) \rightarrow 256 \rightarrow 256 \rightarrow 1$ | $837.0 \pm 43.6$ |
| 75% effects | $(1 \text{ or } 2) \rightarrow 64 \rightarrow 1$ | $343.3 \pm 5.3$ |
| | $(1 \text{ or } 2) \rightarrow 64 \rightarrow 64 \rightarrow 1$ | $422.7 \pm 9.9$ |
| | $(1 \text{ or } 2) \rightarrow 64 \rightarrow 64 \rightarrow 64 \rightarrow 1$ | $506.0 \pm 34.4$ |
| | $(1 \text{ or } 2) \rightarrow 128 \rightarrow 128 \rightarrow 1$ | $552.4 \pm 151.4$ |
| | $(1 \text{ or } 2) \rightarrow 256 \rightarrow 256 \rightarrow 1$ | $1193.8 \pm 0.3$ |

Table F.6: AUC performance with MLP shape functions.

| MLPs with varying complexity | Test AUC |
|---|---|
| $(1 \text{ or } 2) \rightarrow 64 \rightarrow 1$ | $\mathbf{72.90} \pm 0.09$ |
| $(1 \text{ or } 2) \rightarrow 64 \rightarrow 64 \rightarrow 1$ | $72.48 \pm 0.13$ |
| $(1 \text{ or } 2) \rightarrow 64 \rightarrow 64 \rightarrow 64 \rightarrow 1$ | $72.58 \pm 0.17$ |
| $(1 \text{ or } 2) \rightarrow 128 \rightarrow 128 \rightarrow 1$ | $72.34 \pm 0.11$ |
| $(1 \text{ or } 2) \rightarrow 256 \rightarrow 256 \rightarrow 1$ | $71.86 \pm 0.37$ |

the early forward and backward passes of all $p^2$ interaction trees is computationally prohibitive. On the other hand, making this decision prematurely means that the model might not have had enough time to fit terms before suppressing them. We provide an ablation study for model performance as a function of entropy regularization in Table F.7. We observe that when the entropy regularization is

Table F.7: Effect of entropy regularization $\tau$ on test AUC and component selection on Spambase. We report median and mean absolute deviation across 50 runs.

| $\tau$ | 0.00001 | 0.0001 | 0.001 | 0.005 | 0.01 | 0.05 | 0.1 | 1.0 |
|---|---|---|---|---|---|---|---|---|
| AUC | 98.58±0.02 | 98.58±0.02 | 98.54±0.02 | 98.37±0.09 | 98.40±0.12 | 98.40±0.15 | 98.43±0.11 | 98.30±0.08 |
| #Effects Selected | 1653 ± 0 | 1235 ± 164 | 592 ± 173 | 558 ± 328 | 535 ± 352 | 507 ± 451 | 721 ± 528 | 1653 ± 26 |

too high, the model suffers in performance as some informative terms can also get suppressed too quickly. When the entropy regularization is too small, the model can produce very dense solutions, which can hurt interpretability as well as computational scalability. In a reasonable range of entropy regularization, there is a good region where the model produces a sparse solution for high AUC. Interestingly, we also observe that we didn't see a huge sensitivity of variable selection for a range of entropy values in the middle.

## F.6 Effect of screening on performance

Here, we highlight that screening can help improve performance of our model. We consider a challenging Madelon dataset from the NIPS-2003 feature selection challenge, where only a very small subset of features are informative. The number of total features in the dataset are 500. The number of pairwise interactions in this dataset are $\sim 125k$. This is a large combinatorial space for variable selection in terms of interaction effects. A large portion of these interactions are non-informative. Pre-training screening rules similar to the ones used by [34] can effectively reduce the combinatorial space. Recall that the actual number of interactions selected by the model is much smaller than the screened set of interactions through variable selection by our model while training. We can observe from Fig. F.1 that for a range of pre-training screening levels, our model can achieve almost 90% AUC performance — this performance is better than

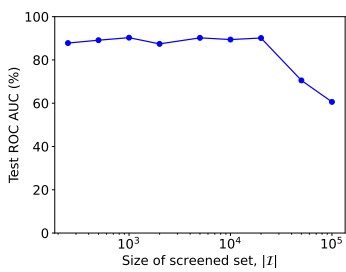

Figure F.1: Performance of GRAND-SLAMIN with various pre-training screening levels on Madelon.

all existing approaches for sparse GAMs with interactions. In particular, the state-of-the art toolkit NODE-GAM that performs end-to-end variable seclection achieves 60% AUC.

Screening also plays a role in faster runtimes. For example, when the screened set is 10,000 interactions out of a total of 125k, the model can be $3\times - 5\times$ faster. Additionally with screening, the overall memory footprint of the model can be much smaller. The overall memory footprint of the model is dictated by the number of interaction effects. With screening, the initial memory footprint of the model can be reduced without a loss in performance.

### F.7 Timing comparison

Our toolkit is highly competitive in terms of training times with all existing tree-based and neural-based toolkits for sparse GAMs with interactions. See Table F.8 for timing comparisons.

Table F.8: Training time in seconds of GRAND-SLAMIN, $EB^2M$, NODE-GA$^2$M, GAMI-Net and SIAN. We report median across 10 runs. Hyphen (-) indicates either the toolkit does not support multiclass e.g., GAMI-Net, SIAN or does not fit interaction effects for multiclass e.g., $EB^2M$.

| Dataset\Model | $EB^2M$ | NODE-GA$^2$M | GAMI-Net | SIAN | GRAND-SLAMIN |
|---|---|---|---|---|---|
| Magic | **140** | 327 | 1567 | 608 | 430 |
| Adult | **284** | 612 | 3611 | 2396 | 1018 |
| Churn | **33** | 699 | 2340 | 298 | 35 |
| Spambase | 297 | 361 | 1979 | 2197 | **133** |
| Miniboone | **1181** | 1523 | 39334 | 64200 | 1662 |
| Online | 402 | **325** | 9030 | 3877 | 432 |
| Bankruptcy | 156 | 240 | 2630 | 3410 | **57** |
| Madelon | 403 | 899 | 5217 | 58500 | **46** |
| Satimage | — | 133 | — | — | **52** |
| Texture | — | 128 | — | — | **98** |
| Optdigits | — | 320 | — | — | **55** |
| Covertype | — | **604** | — | — | 3144 |
| Activity | — | 291 | — | — | **110** |
| Multiple | — | 1417 | — | — | **184** |

## G Additional visualizations for variance reduction in estimation of main effects under structural constraints

Here we consider tract-level American Community Survey dataset from US Census Bureau Planning Database 2022 [55]. Following [23], we consider a reduced dataset with $\sim 39$ covariates (741 possible pairwise interactions) and consider the self-response as the regression target. We fit GRAND-SLAMIN with different structural constraints. We visualize the estimated main effects in the presence of interaction effects for these structural constraints for some features in Figure G.1. We can observe that when additional hierarchy constraints are imposed, the error bars are much smaller across different runs.

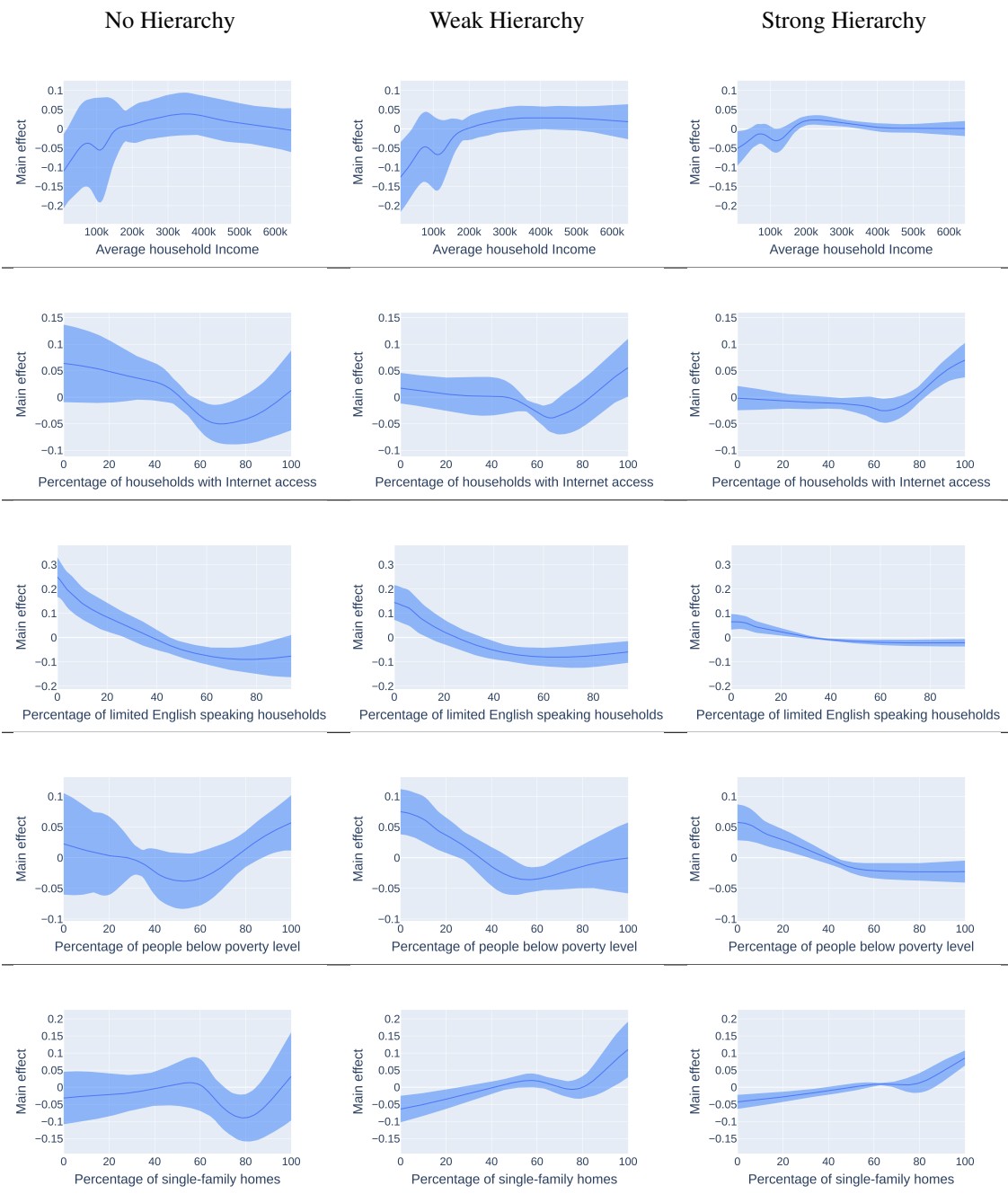

Figure G.1: Estimated main effects in the presence of interaction effects on American Community Survey dataset [Left] without hierarchy, [Middle] with weak hierarchy and [Right] with strong hierarchy. Strong hierarchy has the smallest error bars. We show visualization for 5 features: (a) Row 1: Average household income, (b) Row 2: Percentage of households with internet access, (c) Row 3: Percentage of limited English speaking households, (d) Row 4: Percentage of people below poverty level, (e) Row 5: Percentage of single-family homes.

