# OpenReview forum: "GRAND-SLAMIN’ Interpretable Additive Modeling with Structural Constraints"
_NeurIPS.cc/2023/Conference — NeurIPS 2023 poster_

### Official Review · Reviewer_W32k · 2023-06-18

**Soundness:** 3 good
**Presentation:** 2 fair
**Contribution:** 2 fair
**Rating:** 5
**Confidence:** 4

**Summary:**

This paper introduces hierarchical structure constraints to improve the sparsity of selected features in interaction-aware GAM. The authors use soft relaxations to enable gradient descent during this procedure. Theoretical analyses are given to show statistical properties when using soft tree shape functions in their method.

**Strengths:**

1.	Provides a simple but reasonable design of hierarchical constraints on the sparsity of feature selection in GAM.
2.	Introduces background information and related works in detail.
3.	Offers an efficient implementation that drops unused features during training.

**Weaknesses:**

1.	The technical contribution of this paper is limited. Its framework mainly adopts the existing GA2M with tree-shaped functions. The main contribution is introducing hierarchical constraints to restrict the sparsity of selected features. This is accomplished through learnable indicators that consider the hierarchical constraints. However, this approach is somewhat trivial, as learning Boolean indicators for sparsifying inputs is a common practice. Also, many methods, such as based on Lasso or Gumbel-sigmoid, have been proposed to enable gradient descent on such settings.
2.	The theoretical analysis does not appear to be very relevant to the key contribution claimed. Specifically, the paper discusses the statistical properties of an additive model with soft tree-shaped functions, but does not consider any hierarchy constraints. Since GAM with tree-shaped functions is an existing technique and this paper mainly proposes hierarchy constraints, these analyses do not provide many insights into the methodologies proposed in this paper.
3.	The presentation of this paper is poor. In section 5, many symbols appear without specific explanations, making the analysis confusing and difficult to understand.

**Questions:**

1.	What do $n$, $\bar{u}(f)$, and $\vee $ mean?
2.	What do $a_n$ and $b_n$ represent in the model?
3.	How does the proposed method improve upon other methods that support hierarchy constraints, such as GAMI-Net and SIAN?

**Limitations:**

There is no potential negative societal impact.

---

> ### Author Rebuttal · Authors · 2023-08-10
>
> We thank reviewer for his comments and the time spent in reviewing the paper.
> 1) **Response to Limited Contribution**
>
>   a) **Discussion on GA^2M** We want to clarify that our approach is very different from GA^2M with tree-shaped functions proposed by Lou et al. (2013) and later packaged as EBM. GA^2M is a two-stage approach for the main and interaction effects. In the first stage, they use gradient boosting to fit the main effects---the boosting procedure cycles through all features in a round-robin fashion to fit *all* main effects. In the second stage, they propose a greedy forward stagewise selection strategy to select a small subset of pairwise interactions -- these interaction effects are fitted with shallow tree-like models via gradient boosting. The GA^2M procedure is not based on a joint penalized optimization criterion and, to our knowledge, no statistical guarantees for it are known.
>
> In contrast, the main features of our proposal is to introduce learnable indicator variables to be able to model different structural constraints (e.g., sparse interactions, hierarchy, etc.) Our approach is therefore more flexible. Moreover, ours is an end-to-end optimization framework.
>
>   b) **Discussion on Probabalistic Boolean Indicators, e.g., via Gumbel-sigmoid** Techniques like Gumble-sigmoid have been used in other applications to enforce sparsity. However, in prior GAM with interactions literature, to our knowledge, such stochastic parameterization of boolean indicators have not been used for selection. Moreover, such techniques typically lead to fairly dense solutions because of their probabalistic modeling choice. At inference time, such methods use heuristics via thresholding for actual sparsification. With our Smooth-step approach, our selection is such that the model at training and inference are the same. Additionally, sparsity in Smooth-Step can allow us to successively eliminate the non-selected effects during the course of training, so the time/epoch becomes smaller as epochs progresses, making the algorithm more scalable.
>
>   c) **Discussion/Experiments on Lasso** Lasso is also a possible choice for sparse selection. We compare against a version of Lasso:
> $$min~\hat{E}[l(y, \sum_{i\in[p]}f_i(x_i) + \sum_{i<j}f_{i,j})] + \lambda G,$$
> where $\hat{E}$ denotes the empirical loss on the training dataset, $G=\sum_i||f_i|| + \sum_{i<j} ||f_{i,j}||$ is the group lasso regularization. Overall, our approach consistently outperforms Lasso in terms of AUC performance and variable selection. We provide a subset of results on a few datasets below:
> |       Dataset| Lasso (AUC) | GRAND-SLAMIN    (AUC)         | Lasso   (\#features)     | GRAND-SLAMIN  (\#features) |
> |------------|-------|-------------------------|--------------|--------------|
> | Adult      | 91.19 | **91.54**$\pm0.14$       | 14           | **12**       |
> | Spambase   | 98.32 | **98.81**$\pm0.04$        | 57           | **44**       |
> | Madelon    | 65.22 | **90.13**$\pm1.03$        | 500          | **15**       |
>
> 2) **Theoretical Contribution** To our knowledge, our work is a first to establish theoretical results for learning with soft trees, specifically in the context of GAMs. To construct our theoretical framework, we chose the model with no structural constraints which is one of the main models we discuss. Extending our theoretical framework to include structural constraints is an interesting avenue of research left for future work.
>
> 3)  **Clarification on notations in Section 5** $n$ denotes the number of (training) samples, $\lor$ is used as a shorthand for maximum. $\bar{u}(f)$ can be considered the largest weight (in absolute value) appearing in the model. These will be clarified in the revised version. The sequences $a_n,b_n$ do not necessarily represent any parameters in the model, rather they are constructed as a part of proof (see Lemma B.7 for example). Informally, the fact that $a_n\to 0$ allows us to deduce from Theorem 2 that the error rate $n^{-1/2}$ is achievable as $n\to\infty$.
>
> 4) **GAMI-Net and SIAN vs Our Approach** GAMI-Net and SIAN both are two-stage learning procedures. In the first stage, they use some screening approach to exclusively rely on pre-training screening to select top k important hierarchical interactions. This is a greedy approach. In the second stage, these models fit on all their screened subset. No further selection is performed in this stage.
>
> Our proposal is to introduce learnable indicator variables to be able to model different hierarchy constraints. In contrast to GAMI-Net and SIAN, our approach can learn under such constraints is an end-to-end fashion. Additionally, our approach can also support sparse backpropagation .

---

> > ### Comment · Reviewer_W32k · 2023-08-11
> >
> > Thank you for your response. The authors have partly resolved my concern on the novelty of this paper. While I appreciate the theoretical analysis, I still believe that the method simply incorporates feature selection into a GAM. The soft tree component can be replaced by other implementations. Also, I think the biggest challenge when considering interactions in a GAM is the complexity, and training models for all possible interactions and using sparsification to select them does not seem to be a significant contribution to solving this problem. Since all interaction effects are modeled instead of being heuristically selected, the model complexity at the beginning will be large. This may constrain us to be only able to use shallow or small networks for the major and interaction effects. Indeed, this paper used a shallow model to enable such training, but I doubt its generality in handling sophisticated problems. In practice, some features' effect can be high-order, which may not be captured by the model. As evidence, the adult dataset reported in your paper has an AUC of 91.82. However, I previously tried using NAM with each feature modeled with a network structured as $64 \times 64 \times 64$ and used an embedding layer for categorical features. The AUC can reach 95 without adding any interaction features. Therefore, I will not change my rating.

---

> > > ### Author Response · Authors · 2023-08-17
> > > **Rebuttal by Authors (Part 1)**
> > >
> > > We thank the reviewer for their thoughtful comments. It took us longer than we expected. We were investigating the AUC numbers of adult dataset.
> > > - **Feature/component selection in GAMs** The reviewer’s comment *“simply incorporates feature selection into a GAM”* appears to insinuate that feature/component selection in GAMs with interactions is a trivial problem, which we do not agree with. Feature/Component selection in GAMs with interactions is a challenging combinatorial problem with a long body of prior research [Lou et al. (2013), Yang et al. (2021), Ibrahim et al. (2021), Chang et al. (2022), Enouen and Liu (2022), Radchenko and James (2010), Lin and Zhang (2006)]. Identifying the best subset of components (main and interaction effects) while training is an important and challenging problem. As we discuss in the next point, our joint training (learning components and selection) is very different from approaches such as (i) training all pairwise interactions and then selecting a subset, (ii) selecting a small subset of components prior to training based on some heuristic criteria. Our goal in this paper is to advance the frontiers in this area.  In particular, compared to previous state-of-the-art models, our methodology can provide performance gains. For example, SIAN by Enouen and Liu (2022) purely relies on a pre-training screening approach. We can outperform SIAN in AUC by up to 6% because of our end-to-end optimization approach for component selection. Similarly, EBM by Nori at al. (2019) is not very scalable for datasets with large number of interactions as it follows a greedy forward stagewise selection strategy for identification and fitting of interactions in a boosted fashion. We can outperform EBM for most (11 out of 14) datasets in terms of AUC.
> > > - **Sparse model training can be faster than a dense model** Interestingly, training a model with a few number of components (sparse model) can be faster than training a dense model with all pairwise interactions. This is what we observe even if the nonzero effects are not known a-priori and are learnt during the joint training (learnable indicator variables and components). We view this joint training method as quite different from your described approach *``training models for all possible interactions and using sparsification to select them”*  --- the main reason being that some of the nonzero components are being updated via stochastic gradient descent, and the other components (corresponding to zero indicator variables while training) are effectively removed from training. Additionally, Section 4 (pages 5-6) in the main paper discusses sparse backpropagation tools for speeding up the algorithm. As concrete evidence, we consider the following:
> > >
> > >   - Figure 2 in Section 6.3 in the main paper shows an example where sparse backpropagation leads to $10\times$ speed-up over dense backpropagation.
> > >   - We also show another example where we train a fully dense model with all pairwise interactions ($140,000$) versus our sparse modeling approach with GRAND-SLAMIN. GRAND-SLAMIN selects 587 interactions reaching $230\times$ model compression. We can see $16\times$ faster training times as shown below.
> > >     |Dataset|All pairwise interactions| GRAND-SLAMIN|
> > >     |---|---|---|
> > >     |Activity| ($5182 \pm 16$)sec | (**319** $\pm$ 67)sec |
> > >
> > > References:
> > > - Chun-Hao Chang, Rich Caruana, and Anna Goldenberg. NODE-GAM: Neural generalized additive model
> > > for interpretable deep learning. In ICLR, 2022.
> > > - James Enouen and Yan Liu. Sparse interaction additive networks via feature interaction detection and sparse
> > > selection. In NeurIPS, 2022.
> > > - Shibal Ibrahim, Rahul Mazumder, Peter Radchenko, and Emanuel Ben-David. Predicting Census Survey Response Rates With Parsimonious Additive Models and Structured Interactions. ArXiv, abs/2108.11328, 2021.
> > > - Y. Lin and H. H. Zhang. Component selection and smoothing in multivariate nonparametric regression. The Annals of Statistics, 34:2272–2297, 2006.
> > > - Yin Lou, Rich Caruana, Johannes Gehrke, and Giles Hooker. Accurate intelligible models with pairwise interactions. In KDD, 2013.
> > > - Harsha Nori, Samuel Jenkins, Paul Koch, and Rich Caruana. Interpretml: A unified framework for machine learning interpretability. ArXiv, abs/1909.09223, 2019.
> > > - P. Radchenko and G. M. James. Variable selection using adaptive nonlinear interaction structures in high dimensions. Journal of the American Statistical Association, 105:1541–1553, 2010.
> > > - Zebin Yang, Aijun Zhang, and A. Sudjianto. Gami-net: An explainable neural network based on generalized additive models with structured interactions. Pattern Recognit., 120:108192, 2021.

---

> > > ### Author Response · Authors · 2023-08-17
> > > **Rebuttal by Authors (Part 2)**
> > >
> > > - **We can handle deeper/more complex individual components** The reviewer notes *“This may constrain us to be only able to use shallow or small networks for the major and interaction effects. Indeed, this paper used a shallow model to enable such training, but I doubt its generality
> > > in handling sophisticated problems...“* Thanks for raising this interesting point about complexity of individual components, which we respond to below.
> > >   - **MLPs with varying complexity** We replaced soft trees with MLPs in our framework and did further experiments where we vary the complexity of the individual components. We show training time on News dataset ($p=61$, \# Interactions$=1800$, $N=40000$) below. The results show that the active set (learnt by learnable indicators) in the model is the primary contributing factor in terms of timing. The functions (1 or 2) -> 64 -> 64 -> 1 have at least $22\times$ more parameters than (1 or 2) -> 64 -> 1. However, the time only increases by less than $1.5\times$.
> > > |% Components Selected| MLPs with varying complexity|Training times|
> > > |---|---|---|
> > > | 25%| (1 or 2) -> 64 -> 1| $252.4\pm0.4$|
> > > | 25%| (1 or 2) -> 64 -> 64 -> 1| $247.6\pm0.8$ |
> > > | 25%| (1 or 2) -> 64 -> 64 -> 64 -> 1| $258.8\pm10.7$  |
> > > | 25%| (1 or 2) -> 128 -> 128 -> 1| $268.5\pm19.1$|
> > > | 25%| (1 or 2) -> 256 -> 256 -> 1| $837.0\pm43.6$|
> > > |&nbsp;| |  |
> > > | 75%| (1 or 2) -> 64 -> 1| $343.3\pm5.3$|
> > > | 75%| (1 or 2) -> 64 -> 64 -> 1| $422.7\pm9.9$ |
> > > | 75%| (1 or 2) -> 64 -> 64 -> 64 -> 1| $506.0\pm34.4$|
> > > | 75%| (1 or 2) -> 128 -> 128 -> 1| $552.4\pm151.4$|
> > > | 75%| (1 or 2) -> 256 -> 256 -> 1| $1193.8\pm0.3$|
> > >
> > >     We also show the AUC performance with MLPs with varying complexity below. We did not observe any improvement in
> > >     AUCs with more complex components. It seems to us that since our additive modeling framework is fitting low-dimensional components
> > >      --- 1-dimensional functions for main effects and 2-dimensional functions for pairwise interactions effects --- the complexity of the
> > >     function class doesn’t need to be too large to get good model accuracy.
> > >     |MLPs with varying complexity| Test AUC|
> > >     |---|---|
> > >     | (1 or 2) -> 64 -> 1| **72.90**$\pm0.09$|
> > >     | (1 or 2) -> 64 -> 64 -> 1| $72.48\pm0.13$|
> > >     | (1 or 2) -> 64 -> 64 -> 64 -> 1| $72.58\pm0.17$|
> > >     | (1 or 2) -> 128 -> 128 -> 1| $72.34\pm0.11$|
> > >     | (1 or 2) -> 256 -> 256 ->1| $71.86\pm0.37$|
> > >
> > >     **MLPs or Soft Trees?** We completely agree with your comment *“The soft tree
> > >     component can be replaced by other implementations.“* Interestingly, we did some experiments where we replace soft trees with MLPs
> > >     for individual components. Here MLPs are matched in terms of \#parameters with soft trees.
> > >     | Dataset| GRAND-SLAMIN with MLPs| GRAND-SLAMIN with Soft Trees|
> > >     |---|---|---|
> > >     | Magic| 93.13$\pm$0.12| **93.86**$\pm$0.30|
> > >     | Churn| 92.33$\pm$0.56| **92.40**$\pm$0.41|
> > >     | MiniBooNE| 97.41$\pm$0.21| **97.77**$\pm$0.05|
> > >     | Spambase| 98.27$\pm$0.13| **98.55**$\pm$0.07|
> > >     | News| 72.87$\pm$0.09| **73.24**$\pm$0.04|
> > >
> > >     It seems to us that soft trees seem to have an edge over MLPs across some datasets. But we agree that we can use other components, e.g., MLPs and we will clarify in the revision.
> > >   - **Computational Scalability** As noted above, our method can handle more complex components as long as the number of interactions is not very large. When the number of interactions is quite large (e.g., greater than 100000), we can still handle such problems with some modifications: (i) Sparse training: See point 2 above discussing that sparse training can be faster than fully dense training with all pairwise interactions. (ii) Screening: This is discussed in Section 4 on page 6 in paper. Note that earlier work by  Enouen and Liu (2022) and Yang et al. (2021) also use screening to reduce the number of interaction effects. But there are similarities and differences with what we propose. For example, Yang et al. (2021) use shallow tree-like models to perform screening. However, these methods exclusively rely on screening for component selection. On the other hand, we screen to a much larger set and then perform joint learning of components and learnable indicators on the screened set of interactions to arrive at a better selection of components.
> > >
> > > References:
> > > - James Enouen and Yan Liu. Sparse interaction additive networks via feature interaction detection and sparse
> > > selection. In NeurIPS, 2022.
> > > - Zebin Yang, Aijun Zhang, and A. Sudjianto. Gami-net: An explainable neural network based on generalized additive models with structured interactions. Pattern Recognit., 120:108192, 2021.

---

> > > ### Author Response · Authors · 2023-08-17
> > > **Rebuttal by Authors (Part 3)**
> > >
> > > - **Results on adult dataset** The reviewer’s comment about *“reaching 95% AUC ... with NAM ... without any interactions...“* on adult dataset prompted us to investigate further. We investigated this with 3 different models and all of these approaches seem to reach ~92% test AUC, which is comparable to the number we reported in the paper (We could not get up to 95% test AUC with any of these models). The methods we tried are:
> > >   - NA^2M (NAMs with interactions) from this codebase by Radenovic et al. (2022). This gave a test AUC of 91.3%.
> > >   - LightGBM, which has specific support for categorical variables and models complex interactions between features. We achieved test AUC of 92.3%. We also saw similar test AUC numbers reported by third-party on Kaggle.
> > >   - Our GRAND-SLAMIN framework and modeled as follows: We used a soft tree for each of the effects. For categorical variables, we used embedding layers. We also considered interactions between features and we achieved less than 92% test AUC.
> > >
> > >   Perhaps the reviewer is referring to train AUC and we observe all of these models can reach a train AUC larger than 95%.
> > >
> > > References:
> > > - Filip Radenovic, Abhimanyu Dubey, and Dhruv Mahajan. Neural basis models for interpretability. In NeurIPS, 2022.

---

> > > > ### Comment · Reviewer_W32k · 2023-08-21
> > > >
> > > > Thank you for your detailed rebuttal. I will raise my score to 5.

---

### Official Review · Reviewer_QfPH · 2023-06-26

**Soundness:** 2 fair
**Presentation:** 3 good
**Contribution:** 3 good
**Rating:** 6
**Confidence:** 4

**Summary:**

The present paper presents a method to incprorate pairwise interaction effects into differentiable generalized additive models while retaining scalability.

The authors pose the fitting of the GAM with pairwise interactions as a combinatorial optimization problem with indicator variables and hierarchy constraints added to the pairwise interactions. To be able to efficiently solve this optimization problem, the authors relax the indicator variables using the smooth step function, which is exactly zero or one at the tails of the function, allowing them to discard certain functions during the forward pass. As shape functions, the authors use soft trees, as opposed to e.g. NAMs which use neural networks. To ensure that the gates reach binary state by the end of training they add an entropy regularizer.The authors provide prediction bounds for sparse additive models.

In their experiments, they compare with existing method such as Node-GAM which incorporate pairwise interactions and show favourable results.

**Strengths:**

- The paper is overall well written and the structure is easy to follow.
- The methodological contribution is neat and only adds few hyperparameters.
- The proposed method produces sparser pairwise interactions which are also faster to train than the compared methods.

**Weaknesses:**

- Table 2, Table 3, Table 4, Figure 2 please provide the number of runs/seeds and either the standard error, mean absolute deviation or bootstrap confidence intervals for the results. The present analysis could be misleading if it was only carried out over one seed.
- As your method is applicable to any shape function class, it would be interesting to show results for shape functions like in Neural Additive Models modelled via MLPs.
- I would have expected the models with weak hierarchy to strictly outperform the models with strong hierarchy. Could you comment on this?

Typos:
- Line 26 'xchallenging'
- Line 46 'pose'
- Line 87 'first step'
- Line 328 'significantly/much faster'


Improvements:
- Consider increasing the font size of Figure 1 and Figure 2.

**Questions:**

- I think the contribution of the paper is sound, but currently their experimental evaluation is weak as the comparison between methods appears to be carried out on a single seed (I saw some of the results in the appendix have results over additional seeds, but these results need to be in the main paper.). I would consider increasing my score if this is addressed.

- How sensitive is the variable selection to the entropy regularization strength?

**Limitations:**

- Limitations are not discussed in the paper.

---

> ### Author Rebuttal · Authors · 2023-08-10
>
> We thank reviewer for his comments and the time spent in reviewing the paper.
>
> 1) **Multiple runs across seeds** Tables F.1 and F.2 in the Supplement correspond to 10 runs/seeds for the optimal trials reported in Table 2 and Table 3. We will use F.1 and F.2 in the main text in the camera-ready version.
>     We will also replace Table 4 with it's multiple seeds version in the camera-ready draft.
>
> 2) **Comparison with MLP shape functions** We have performed new experiments for GRAND-SLAMIN with MLP shape functions.
>     For a fair comparison with Soft trees, we used MLPs, which had similar number of parameters as depth-4 soft trees. We observed soft trees  to be more expressive when parameter-matched as shown below.
> | Dataset   | GRAND-SLAMIN with MLP basis | GRAND-SLAMIN with Soft Trees basis |
> |-----------|-----------------------------|------------------------------------|
> | Magic     | 93.18                       | **94.48**                          |
> | Adult     | 90.03                       | **91.81**                          |
> | Churn     | 92.55                       | **93.88**                          |
> | MiniBooNE | 97.48                       | **97.85**                          |
> | Spambase  | 98.16                       | **98.64**                          |
> | News      | 73.19                       | **73.29**                          |
>
> 3) **Weak hierarchy vs strong hierarchy** It could be expected that models with hierarchy generally may outperform models with strong hierarchy as the former imposes lesser restrictions. The reviewer can refer to tables F.1 and F.2 in the Supplement (which provide median numbers when the models are run with different seeds). Indeed, typically, models with weak hierarchy tend to outperform models with strong hierarchy. However, there are some 1-2 datasets where strong hierarchy still outperforms in terms of performance in comparison with weak hierarchy.
>
>     However, strong hierarchy has other advantages. Models with strong hierarchy are considered to be more interpretable. They also lead to more compact feature selection as shown in Table 4. Models with lesser number of features are easier to interpret.
>
> 4) **Typos and Figure fonts** Thanks for bringing this to our attention. Typos are addressed and font size of Figures 1 and 2 will be increased for the camera-ready revision.
>
> 5) **Sensitivity to Entropy regularization** We provide an ablation study on Spambase for effect of entropy regularization on model performance and variable selection. Please see Figure R2 in figure rebuttal document. For each entropy regularization, we select the best model corresponding to the optimal selection regularization ($\lambda$) based on a validation set. Overall, we see small effect of entropy regularization on variable selection and very little effect on test AUC.

---

> > ### Comment · Reviewer_QfPH · 2023-08-10
> > **Thank you for your detailed answer**
> >
> > Thank you for your detailed answers to all of our reviews.
> >
> > Concerning my review:
> > 1. Noted.
> > 2. Thank you for carrying out the extra experiment. Please update the table with the mean absolute deviations or bootstrap confidence bounds. Over how many seeds did you evaluate and how did you optimize the hyperparameters of the neural network? What search space did you use for the hyperparameter optimization?
> > 3. Thank you for your detailed answer.
> > 4. Noted.
> > 5. Thank you for providing the extra figures. I believe the authors are referring to Figure 1 not 2 in the rebuttal document. It is interesting to see that entropy regularization has no significant effect in the range you investigated. How many seeds did you use for each $\tau$? Could you increase the range and report when the effect of $\tau$ becomes noticeable? Please also adapt the font size as it is too small.
> >
> > Concerning other reviews:
> > - Thank you for providing the shape functions shown in Figure 2 of the rebuttal pdf. It's nice to see that the variance decreases when adding the hierarchical constraints.

---

> > > ### Author Response · Authors · 2023-08-17
> > > **Rebuttal by Authors**
> > >
> > > We thank the reviewer for their insightful comments and questions.
> > > - Here is the updated table with median and mean absolute deviations across 10 runs for MLP and Soft Trees:
> > >
> > >     | Dataset   | GRAND-SLAMIN with MLPs | GRAND-SLAMIN with Soft Trees |
> > >     |-----------|------------------------|------------------------------|
> > >     | Magic     | 93.13$\pm$0.12         | **93.86**$\pm$0.30           |
> > >     | Adult     | 90.79$\pm$0.05         | **91.54**$\pm$0.14           |
> > >     | Churn     | 92.33$\pm$0.56         | **92.40**$\pm$0.41           |
> > >     | MiniBooNE | 97.41$\pm$0.21         | **97.77**$\pm$0.05           |
> > >     | Spambase  | 98.27$\pm$0.13         | **98.55**$\pm$0.07           |
> > >     | News      | 72.87$\pm$0.09         | **73.24**$\pm$0.04           |
> > >
> > >     We considered two NN shape functions: (1 or 2) -> 16 -> 1 and  (1 or 2) -> 8 -> 4 ->1. We used grid search for tuning over the two
> > >     architectural choices of NN shape functions, and all the other model and optimizer related hyperparameters: learning rate $\in ${0.05,
> > >     0.01, 0.005}, batch-size $\in$ {64,256}, $\lambda \in$ {0, $10^{-6}$, $10^{-5.6}$, $\cdots$, $10^{-3.3}$, $10^{-3}$} for selection
> > >     regularization, $\gamma \in$ {0.01, 0.1, 1} for Smooth-step, $\tau \in ${0.01, 0.1, 1.0} for entropy regularization. We also tried all 3 types
> > >     of structural constraints.
> > >
> > > - In the range we highlighted, we didn’t see a huge impact to show the model is not highly sensitive to the values of $\tau$ in reasonable range. When we use overly small or overly large values, we do see a big impact on variable selection. Here are the median numbers along with the mean absolute deviations across 50 runs for different values of $\tau$.
> > > | &nbsp;&nbsp;&nbsp;&nbsp;&nbsp;&nbsp;&nbsp;&nbsp;&nbsp;&nbsp;&nbsp;&nbsp;&nbsp; $\tau$ | &nbsp;&nbsp;&nbsp;0.00001       | &nbsp;&nbsp;&nbsp;&nbsp;&nbsp;0.0001        | &nbsp;&nbsp;&nbsp;&nbsp;&nbsp;0.001         | &nbsp;&nbsp;&nbsp;&nbsp;&nbsp;&nbsp;0.01          | &nbsp;&nbsp;&nbsp;&nbsp;&nbsp;&nbsp;&nbsp;0.1           | &nbsp;&nbsp;&nbsp;&nbsp;&nbsp;&nbsp;&nbsp;&nbsp;1.0           | &nbsp;&nbsp;&nbsp;&nbsp;&nbsp;&nbsp;&nbsp;10.0          |
> > > |--------------------|----------------|----------------|----------------|----------------|----------------|----------------|----------------|
> > > | &nbsp;&nbsp;&nbsp;&nbsp;&nbsp;&nbsp;&nbsp;&nbsp;&nbsp;&nbsp;&nbsp;AUC               | 98.58$\pm$0.02 | &nbsp;98.58$\pm$0.02 | 98.54$\pm$0.02 | 98.40$\pm$0.12 | 98.43$\pm$0.11 | 98.30$\pm$0.08 | 98.29$\pm$0.09 |
> > > | &nbsp;\#Effects selected | &nbsp;&nbsp;$1653\pm0$    | $1235\pm164$  | &nbsp;$592\pm173$   | &nbsp;$535\pm352$   | &nbsp;$721\pm528$   | &nbsp;$1653\pm26$   | &nbsp;$1653\pm27$   |
> > > - We will increase the font size for the figures.

---

> > > > ### Comment · Reviewer_QfPH · 2023-08-21
> > > >
> > > > I would like to thank the authors for the thorough rebuttal and the extra experiment evaluating the range of $\tau$.
> > > >
> > > > My concerns have been addressed and I will raise my score to 6.

---

### Official Review · Reviewer_xnEj · 2023-07-06

**Soundness:** 3 good
**Presentation:** 3 good
**Contribution:** 3 good
**Rating:** 6
**Confidence:** 3

**Summary:**

The authors introduce GRAND-SLAMIN’, a framework for end-to-end learning of sparse Generalized Additive Models (GAMs). GRAND-SLAMIN’ utilizes soft neural decision trees as base learners and incorporates support for second-order interactions and structural constraints. Sparse variable selection is achieved via a smooth relaxation of binary gate variables which play the role of suppressing unused terms. These gates can also be combined to enable structural constraints, restricting second-order interaction to participating first-order terms. This formulation facilitates a sparse backpropagation strategy in which suppressed terms and interactions are pruned from the model early in training. The experiments demonstrate promising results in terms of prediction accuracy and scalability. The paper also establishes novel non-asymptotic prediction bounds for GAM estimators with soft tree-based shape functions, providing a theoretical foundation for their approach.

**Strengths:**

* The end-to-end differentiable parameterization for feature selection is a novel and valuable contribution. It greatly simplifies the implementation of the feature selection and feature interaction detection procedure and integrates well within the model optimization process. It also enables the enforcement of structural constraints, further enhancing the flexibility and interpretability of the framework.

* The inclusion of non-asymptotic bounds provide a theoretical basis for the behavior and performance of the proposed approach, showcasing that robustness to noise improves as more samples are observed.

* The experimental results also support the intuition that incorporating feature selection into the Generalized Additive Model (GAM) can benefit robustness to noise and its generalization. This finding is interesting and can maybe be linked to previous work on feature selection and robustness: L1 regularized regression (Lasso) is a form of robustness to feature-wise disturbance [Xu, 2008] and selecting relevant subsets of features in feature bagging is known to help model performance [Tian, 2021].

[Xu, 2008]: Xu, H., Caramanis, C., & Mannor, S. (2008). Robust Regression and Lasso. Advances in Neural Information Processing Systems, 21. [https://proceedings.neurips.cc/paper_files/paper...](https://proceedings.neurips.cc/paper_files/paper/2008/hash/24681928425f5a9133504de568f5f6df-Abstract.html)\
[Tian, 2021]: Tian, Y., & Feng, Y. (2021). RaSE: Random Subspace Ensemble Classification. Journal of Machine Learning Research, 22(45), 1–93.
https://jmlr.org/papers/v22/20-600.html

**Weaknesses:**

* The lack of visualizations of local and global explanations which can be generated with the proposed framework hinders the assessment of interpretability. Visualizations play an important role in presenting and communicating the decision-making process of interpretable models and shedding light on the contribution of individual features and their interactions.

* The binary gates snap into place reaching a permanent state of 0 or 1 as training progresses. This is understandably a desired property of smoothstep activation, however it means that the model cannot update decisions regarding the usefulness of input features once this permanent state is reached.

**Questions:**

* I think the inclusion of both local and global visualizations of the recovered shape functions would significantly improve the clarity and impact of the paper. Illustrating the neural soft decision trees would help readers gain a deeper understanding of how the input features contribute to model predictions and encourage its adoption down the line.

* My intuition leads me to believe that there might be a trade-off in the speed at which the binary state of the gate variables should be reached. On the one-hand suppression of the un-informative terms should happen fast as the early forward and backward passes of all $p^2$ interaction trees is computationally prohibitive. On the other, making this decision prematurely means that the model might not have had enough time to fit terms before suppressing them. Maybe you could observe as an ablation model performance as a function of the entropy regularizer hyperparameter $\tau$?

* Related to the above point, do you find that models with different random initializations agree on the usefulness of features, meaning, are the selected features and selected interactions consistent between runs? Perhaps there is a metric which can quantify this (for ex. Jaccard similarity) and you could numerically compare with the screening approach of the other baselines.

* I am not familiar with neural soft decision trees but they seem like they could be quite sensitive to the choice of hyperparameters, and in particular the choice of learning rate. How difficult are they to train in practice? Can you provide some details on the Optuna hyperparameter search: Is it very time consuming? Is the approximated hyperparameter objective function smooth?



**Limitations:**

No, I think the limitations could be better addressed by dedicating a section in the main text.

---

> ### Author Rebuttal · Authors · 2023-08-10
>
> We thank reviewer for his comments and the time spent in reviewing the paper.
>
> 1) **Visualization** We provide a new visualization study to further highlight an important contribution of our paper.
>     In particular, our framework can support models with structural constraints. Hence, we study the effect of these constraints on the stability of learning main effects (in the presence of interactions) when these structural constraints are imposed or not. For this exercise, we consider bikesharing dataset. We visualize some of the main effects in the presence/absence of hierarchy in Figure
>     R1 in rebuttal figure document. We can observe that when additional hierarchy constraints are imposed, the error bars are much more compact across different runs. This can potentially increase the trust you can have on the model for deriving interpretability insights.
>     The figures also highlight that soft trees are smooth basis functions for learning effects.
>
> 2) **Speed of reaching binary states** It is correct that reaching the binary state should not be premature. For this reason, our learnable indicator variables are initialized to be around $0.5$. We also do not impose a very large entropy regularization. This gives the algorithm sufficient time to learn the informative terms. Additionally, a smaller learning rate choice can also be useful to give the model more time to find the informative terms.
>
> 3) **Effect of entropy on performance and variable selection** We provide an ablation study on Spambase for effect of entropy regularization on model performance and variable selection. Please see Figure R2 in figure rebuttal document. For each entropy regularization, we select the best model corresponding to the optimal selection regularization ($\lambda$) based on a validation set. Overall, we see small effect of entropy regularization on variable selection and very little effect on test AUC.
>
> 4) **Sensitivity to hyperparameters** We observed soft trees to be generally not very sensitive to the choice of typical hyperparameters, e.g., batch size and learning rates. In our tuning, we only considered 3 learning rates $\{0.05, 0.01, 0.005\}$ and 2 different batch sizes $\{64,256\}$. However, the most important parameter for our modeling framework was $\lambda$ for sparsity and hence we tuned over a wide range of 11 $\lambda$ values in the set $(0, 1e-6, \cdots, 1e-3)$. We can highlight this more in the camera-ready version. More information about the hyperparameter tuning with Optuna are included in the Supplement.

---

> > ### Comment · Reviewer_xnEj · 2023-08-14
> >
> > (1). Thank you, the visualizations you provided are interesting and do go in the direction of improving the paper. Would be nice to see this for more features/datasets in a dedicated section of the appendix.  May I ask how many runs were used to generate the error bars? Are these generated using standard error (SE) across runs?
> >
> > (2), (3) and (4). Figure R1 does appear to agree with your points here. However, I find it rather surprising that the model performance does not change as you vary $\tau$. What happens when you use overly large or overly small values? I agree with reviewer QfPH that you should increase the range of values so that it is clear if/when performance starts degrading. Would have been nice to display error bars for this figure too.

---

> > > ### Author Response · Authors · 2023-08-17
> > > **Rebuttal by Authors**
> > >
> > > We thank the reviewer for their insightful comments and questions.
> > > 1. We will include visualizations for more features in a dedicated section in the Appendix. We used 10 runs to generate the error bars and we used mean absolute deviation (MAD) to generate the error bars across the runs.
> > >
> > > 2. In the range we highlighted, we didn’t see a huge impact to show the model is not highly sensitive to the values of $\tau$ in a reasonable range. When we use overly small or overly large values, we do see a big impact on variable selection. Here are the median numbers along with the mean absolute deviations across 50 runs for different values of $\tau$.
> > >
> > > | &nbsp;&nbsp;&nbsp;&nbsp;&nbsp;&nbsp;&nbsp;&nbsp;&nbsp;&nbsp;&nbsp;&nbsp;&nbsp; $\tau$ | &nbsp;&nbsp;&nbsp;0.00001        | &nbsp;&nbsp;&nbsp;&nbsp;&nbsp;0.0001         | &nbsp;&nbsp;&nbsp;&nbsp;&nbsp;0.001          | &nbsp;&nbsp;&nbsp;&nbsp;&nbsp;&nbsp;0.01           | &nbsp;&nbsp;&nbsp;&nbsp;&nbsp;&nbsp;&nbsp;0.1            | &nbsp;&nbsp;&nbsp;&nbsp;&nbsp;&nbsp;&nbsp;&nbsp;1.0            | &nbsp;&nbsp;&nbsp;&nbsp;&nbsp;&nbsp;&nbsp;10.0           |
> > > |--------------------|----------------|----------------|----------------|----------------|----------------|----------------|----------------|
> > > | &nbsp;&nbsp;&nbsp;&nbsp;&nbsp;&nbsp;&nbsp;&nbsp;&nbsp;&nbsp;&nbsp;AUC                | 98.58$\pm$0.02 | &nbsp;98.58$\pm$0.02 | 98.54$\pm$0.02 | 98.40$\pm$0.12 | 98.43$\pm$0.11 | 98.30$\pm$0.08 | 98.29$\pm$0.09 |
> > > | &nbsp;\#Effects selected | &nbsp;&nbsp;$1653\pm0$     | $1235\pm164$   | &nbsp;$592\pm173$    | &nbsp;$535\pm352$    | &nbsp;$721\pm528$    | &nbsp;$1653\pm26$    | &nbsp;$1653\pm27$    |

---

### Official Review · Reviewer_WskZ · 2023-07-07

**Soundness:** 3 good
**Presentation:** 3 good
**Contribution:** 3 good
**Rating:** 6
**Confidence:** 4

**Summary:**

The authors proposed an GAM under structure constraints. By incorporating herarical constraint, the proposed method presents better performance than two stage methods. Theoritical analysis are also provided to bound the generalization error.

**Strengths:**

A GAM method was proposed, which can be learnt in an end to end manner. By introducing several binary variables, herachical constraints are automatically imposed on the model.
The theoritical parts are new.

**Weaknesses:**

1, Although the authors claim their model can be learnt efficienctly, there still is p^2 models need to be learnt, which is unpractical for large scale datasets.
2, There is no explanation for why soft trees are used. Actually, the soft tree can be viewed as a NN with special    structure.  So why MLP is not used?
3, Although same hierarchy structures are also used in GAMI-NET, nevertheless, why it helps to improve the performance of the model is still unclear to me. The motivation to use such a structure is weak in this paper.

**Questions:**

1, which screening method is used in your method? Does  it same to that in EBM?

**Limitations:**

yes

---

> ### Author Rebuttal · Authors · 2023-08-10
>
> We thank reviewer for his comments and the time spent in reviewing the paper.
>
> 1) **Sparsity** The goal of our method is to recover a subset of $p^2$ effects. We achieve this goal with two ideas.
> For small to medium sized datasets, we recover an active subset via an end-to-end learning optimization procedure.
> For large-scale datasets, we perform a screening to reduce the initial combinatorial search space from $p\choose2$ to e.g., 5000 or 10000. Later our end-to-end optimization on this screened set further estimates a much sparse set. Hence the model doesn’t estimate $p^2$ trees. It has been highlighted in prior literature (Nori et al. (2019), Ibrahim et al. (2021)) that using all pairwise interactions is unnecessary, can hurt statistical performance and makes interpretability more challenging. In all cases, the training can be efficiently performed using sparse backpropagation. Sparsifying the model during training makes our approach scalable for settings with large number of interactions.
>
> 2) **Soft Trees vs MLP shape functions** Our use of soft trees was primarily motivated by their good performance despite being compact structures. To highlight why Soft trees were selected as opposed to MLPs, we performed new experiments where our models were estimated with MLP shape functions. For a fair comparison, we used MLPs, which were matched in parameters to depth-4 soft trees. We observed, soft trees seemed to be more expressive as shown below.
> | Dataset | GRAND-SLAMIN with MLP basis | GRAND-SLAMIN with Soft Trees basis |
> |------------|----------------------------|-----------------------------------|
> | Magic      | 93.18                      | **94.48**                    |
> | Adult      | 90.03                      | **91.81**                   |
> | Churn      | 92.55                      | **93.88**                    |
> | MiniBooNE  | 97.48                      | **97.85**                   |
> | Spambase   | 98.16                      | **98.64**                   |
> | News       | 73.19                      | **73.29**                    |
>
> 3) **Motivation of Hierarchy Structure**
>     The motivation to use the structural constraints is multi-fold (i) better interpretability (Bien et al. (2013), Enouen and Liu (2022) ) (ii)  more compact feature selection as shown in Table 4. Lesser number of features also improve interpretability (iii) lesser variance in estimation of main effects (in the presence of interaction effects) --- see visualizations (Figure R1) in the global rebuttal document. This allows the user to have more trust on model interpretability explanations.
>
>     Generally speaking, additional constraints can help regularize a model or reduce variance, sometimes resulting in improved AUC.
>     Since these are real datasets, we are unable to make definitive statements about the structure of a *true* underlying model.
>
> 4) **Method of Screening** For screening on large-scale datasets, we used CART for each pairwise interaction and sorted the interaction effects based on AUC performance to select a screened set of interaction effects. Note that this is done only at the start of training. For example, on Madelon dataset. this screening took 30 second to fit CART for all of the 125000 interaction effects separately in a multithreaded environment.
>     The selection method in EBM also used shallow tree-like models to select the interaction effects. However, note that EBM uses their screening approach in their greedy forward stagewise selection to select the best interaction effect in their boosting algorithm.
>
> References:
> 1) Jacob Bien, Jonathan Taylor, and Robert Tibshirani. A lasso for hierarchical interactions. The Annals of Statistics, 41(3):1111–1141, June 2013.
> 2) James Enouen and Yan Liu. Sparse interaction additive networks via feature interaction detection and sparse selection. In Alice H. Oh, Alekh Agarwal, Danielle Belgrave, and Kyunghyun Cho, editors, Advances in Neural Information Processing Systems, 2022.
> 3) Harsha Nori, Samuel Jenkins, Paul Koch, and Rich Caruana. Interpretml: A unified framework for machine learning interpretability. ArXiv, abs/1909.09223, 2019.
> 4) Shibal Ibrahim, Rahul Mazumder, Peter Radchenko, and Emanuel Ben-David. Predicting census survey response rates via interpretable nonparametric additive models with structured interactions, 2022.

---

### Official Review · Reviewer_nF9H · 2023-07-12

**Soundness:** 3 good
**Presentation:** 3 good
**Contribution:** 4 excellent
**Rating:** 6
**Confidence:** 4

**Summary:**

This paper deals with interpretable machine learning. Authors propose a new interpretable model based on generalized additive modeling. Namely they propose a scalable solution for sparse interaction modeling. They compare against related work on interpretable benchmarks.

**Strengths:**

+ Novel contributions: novel optimization of additive models with indicator variables; backward pass customization for indicator functions.
+ Really good theoretical analysis that is greatly appreciated, especially the level of detail in the appendix
+ Well and clearly explained approach
+ SOTA performance on a wide range of datasets
+ Very scalable as shown with detailed timing results from the appendix

**Weaknesses:**

- Somewhat weird choice of datasets. Eg, some standard datasets from papers like NAM, NBM, SPAM, NODE-GAM missing. Most of the datasets presented in this work have a very small difference in performance between the methods, more or less they all perform the same.
- EBM also has EB2M version, unclear why is it not compared with it, or am I wrongly understanding the tables? Some strong related works with open source libraries missing: NAM, NBM, SPAM, all three are available with 2nd and some even with 3rd order interactions.
- Unclear why would a use of such a model select no hier, weak hier, or strong hier. Depending on the dataset, different hierarchy performs the best.
- Can this approach model 3rd order interactions, it would be very interesting to see results on that?
- Is there any interpretability difference between this model and eg NODE-GAM at 2nd order? Is there any benefit in such a sparsity constrains and trainable selection functions over NODE-GAM? Analysis showing some insight into this would improve the paper.

**Questions:**

See points in weaknesses.

**Limitations:**

Non adequate limitations discussion in the main paper.

---

> ### Author Rebuttal · Authors · 2023-08-10
>
> We thank reviewer for his comments and the time spent in reviewing the paper.
>
> 1) **Choice of datasets** A central theme of our work is a sparse selection of main/interaction effects. Our selection of datasets was motivated to consider tabular classification datasets which have a sufficient number of possible pairwise interaction effects. Most of the tabular classification datasets e.g., in NODE-GAM paper consider datasets with $p\leq30$ (\#interactions $\leq \sim 400$). This is the main motivation to consider a different set of datasets so that sparsity can play a more important role when the number of interactions are not too small. Additionally, the differences in performance across methods is statistically significant --- please see the mean absolute deviations in Tables F.1 and F.2 in our Supplemental document.
>
> 2) **EB^2M notation** We used EBM with interactions. We will rename it to  EB^2M.
>
> 3) **Comparison with NA^2M, NB^2M and SPAM** NA^2M, NB^2M and SPAM appear to be dense approaches that fit all pairwise interactions. Hence, we compared only with methods that learn sparse interactions. However, based on your comment, we have run new experiments to provide comparisons with NA^2M, NB^2M and SPAM. The performance numbers are given below:
> | Dataset      | NA^2M (Dense) | NB^2M (Dense) | SPAM (Dense) | GRAND-SLAMIN (Sparse)|
> |--------------|---------------|---------------|--------------|----------------------|
> | Magic        | 94.27         | 93.76         | 91.54        | **94.48**            |
> | Adult        | 91.33         | 91.28         | 89.89        | **91.81**            |
> | Churn        | 92.81         | 92.52         | 87.10        | **93.88**            |
> | Satimage     | 98.78         | **98.92**     | 97.73        | 98.84                |
> | Covertype    | 98.16         | **98.44**     | 96.21        | 98.40                |
> | Spambase     | 98.41         | 98.43         | 97.81        | **98.64**            |
> | News         | 71.62         | 71.41         | 70.72        | **73.29**            |
> | Bankruptcy   | **94.01**     | 90.36         | 86.95        | 93.01                |
>
>     For larger datasets e.g., Madelon, Activity and Multiple, these approaches ran out-of-memory on a single V100 Tesla GPU. We generally outperform all these 3 approaches in terms of performance. We also support sparsity and hierarchy for easier interpretability.
>
> 3) **No hierarchy vs hierarchy** Generally speaking, additional constraints can help regularize a model or reduce variance, sometimes resulting in improved AUC.
>     Since these are real datasets, we are unable to make definitive statements about the structure of a `true’ underlying model. In practice, the model with the best predictive performance or better interpretability can be chosen depending on context.
>
> 4) **3rd order interactions** Our approach can be extended to model 3rd order interactions.
>     In the case of third-order interactions, the structural constraints for strong and weak hierarchy can be defined as described next. For strong hierarchy, if we consider the constraint $f_{i,j,k}\neq 0 \Longrightarrow f_i \neq 0\text{ and }f_j \neq 0 \text{ and } f_k \neq 0$, this can be modeled as:
>     \begin{align}
>         q(z_i,z_j,z_k,z_{i,j,k})=z_iz_jz_kz_{i,j,k}
>     \end{align}
>     For weak hierarchy, if we consider the constraint $ f_{i,j,k}\neq 0 \Longrightarrow f_i \neq 0\text{ or }f_j \neq 0 \text{ or } f_k \neq 0$, we can model it as:
>     \begin{align}
>         q(z_i,z_j,z_k,z_{i,j,k})=(z_i + z_j + z_k - z_iz_j - z_iz_k - z_jz_k + z_iz_jz_k) z_{i,j,k}
>     \end{align}
>     Hence, our sparse selection and hierarchy constraints can add value in terms of model compactness and feature selection in the settings with third-order interactions as well.
>     However, for third-order interactions, the number of third-order interactions can be of the order $~p^3$.
>     Hence, for scalability, this would also require using a pre-training screening approach.
>     Despite the fact that our approach is generalizable to third-order interactions, we do not consider such interactions because these are not considered to be easily interpretable.
>
> 5) **NODE-GAM vs Our Approach** There are differences in modeling choices by NODE-GAM and our approach. NODE-GAM parameterizes each tree as a NODE tree using all features. Each component learns to use at most 2 features by the end of training (via the entmax transformation). Sparsity in NODE-GAM is achieved by constraining the number of trees. In contrast, our approach explicitly models the pairwise interactions with soft trees. Each tree is explicitly modeled to rely on a pair of features (using lesser number of parameters than NODE trees). The optimization framework uses trainable selection functions $z$ to selects a subset of trees.
>
>     We also show how additional constraints on $z$ can be imposed to achieve hierarchy (weak or strong).
>     It is unclear how NODE-GAM can be generalized to support hierarchy constraints. These constraints can add more meaningful interpretability and improve variable selection without significantly impacting model performance.

---

### Author Rebuttal · Authors · 2023-08-10

We thank all reviewers for their comments and the time spent in reviewing the paper.

Please see our attached figure rebuttal document.

---

### Decision · Program_Chairs · 2023-09-21

**Decision:**

Accept (poster)

**Comment:**

The reviewers praise the novelty of the approach, the theoretical analysis (esp. non-asymptotic results), the empirical results, the scalability, and the end-to-end trainability. The main criticisms are the choice of benchmark datasets, the lack of some baselines, the applicability to third-order interactions, the motivation for the hierarchical approach, lack of visualizations, and sensitivity to hyperparameter choices. However, most, if not all, of these concerns have been addressed by the rebuttal.